# Why Settle for Mid: A Probabilistic Viewpoint to Spatial Relationship Alignment in Text-to-image Models

**Parham Rezaei**                                                            *parhamix@gmail.com*
*Department of Computer Engineering*
*Sharif University of Technology*

**Arash Marioriyad**                                                     *arashmarioriyad@gmail.com*
*Department of Computer Engineering*
*Sharif University of Technology*

**Mahdieh Soleymani Baghshah**                                          *soleymani@sharif.edu*
*Department of Computer Engineering*
*Sharif University of Technology*

**Mohammad Hossein Rohban**                                               *rohban@sharif.edu*
*Department of Computer Engineering*
*Sharif University of Technology*

**Reviewed on OpenReview:** *https://openreview.net/forum?id=mFlanJKVFD*

## Abstract

Despite the ability of text-to-image models to generate high-quality, realistic, and diverse images, they face challenges in compositional generation, often struggling to accurately represent details specified in the input prompt. A prevalent issue in compositional generation is the misalignment of spatial relationships, as models often fail to faithfully generate images that reflect the spatial configurations specified between objects in the input prompts. To address this challenge, we propose a novel probabilistic framework for modeling the relative spatial positioning of objects in a scene, leveraging the concept of *Probability of Superiority (PoS)*. Building on this insight, we make two key contributions. First, we introduce a novel evaluation metric, *PoS-based Evaluation (PSE)*, designed to assess the alignment of 2D and 3D spatial relationships between text and image, with improved adherence to human judgment. Second, we propose *PoS-based Generation (PSG)*, an inference-time method that improves the alignment of 2D and 3D spatial relationships in T2I models without requiring fine-tuning. *PSG* employs a PoS-based reward function that can be utilized in two distinct ways: (1) as a *gradient-based* guidance mechanism applied to the cross-attention maps during the denoising steps, or (2) as a *search-based* strategy that evaluates a set of initial noise vectors to select the best one. Extensive experiments demonstrate that the *PSE* metric exhibits stronger alignment with human judgment compared to traditional center-based metrics, providing a more nuanced and reliable measure of complex spatial relationship accuracy in text-image alignment. Furthermore, *PSG* significantly enhances the ability of text-to-image models to generate images with specified spatial configurations, outperforming state-of-the-art methods across multiple evaluation metrics and benchmarks.[1]

## 1 Introduction

Recent advancements in computational power and large multimodal datasets have led to the development of powerful text-to-image (T2I) models, such as Stable Diffusion (2; 1) and DALL-E (3; 4), which generate high-

---

[1]The codebase is available at `https://github.com/Rezaei-Parham/Probabilistic-Spatial-Alignment`.

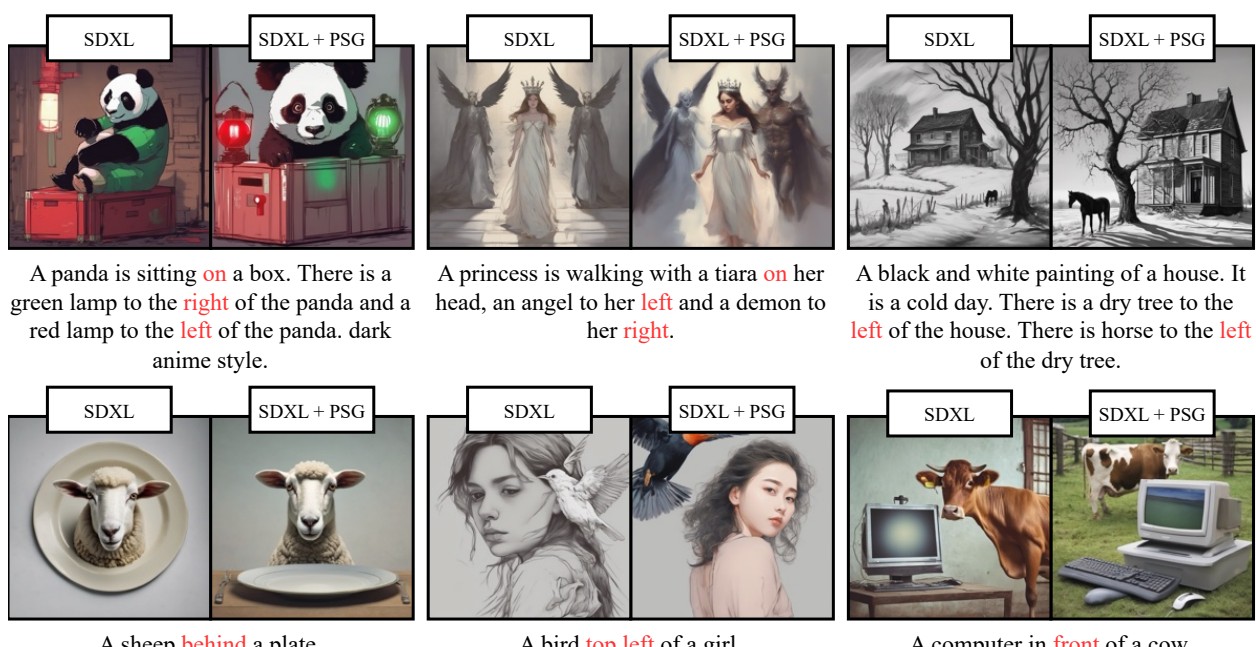

Figure 1: Qualitative comparison of our proposed method, *PSG*, using SDXL (1) as the backbone with the baseline on a variety of complex prompt with one or more 2D and 3D spatial relationships. Our method generates images with accurate spatial relations even for prompts defining several relations between objects.

quality and diverse images from text descriptions. However, concerns have arisen regarding their reliability in generating images that align with the specific details in input prompts, leading to well-known failures referred to as compositional generation issues.

These failures can be categorized into four main types (5): *object missing*, where the generated image omits one or more objects explicitly mentioned in the prompt (6; 7; 8; 9); *incorrect attribute binding*, where attributes are not correctly associated with entities (10; 11; 12); *incorrect spatial relationships*, where the spatial arrangement of objects in the image does not match the description (13; 14); and *numeracy-related* issues, where the model fails to represent the exact specified number of entities in the prompt (15; 16). This work addresses the challenge of 2D and 3D spatial relationships in T2I models, focusing on overcoming two key hurdles: generating all specified objects without omission and demonstrating a robust geometric understanding to accurately interpret relative positions and orientations.

Improving compositional generation in T2I models can be achieved through fine-tuning on large multimodal datasets (17; 14; 18), a process that is costly, time-consuming, and infeasible in resource-limited settings. Alternatively, training-free methods (19; 12; 6; 11; 10; 7; 20; 21) have gained popularity for adjusting latent embeddings during inference without modifying model parameters, thereby preserving model integrity and preventing overfitting. Another approach involves incorporating supplementary inputs, such as layouts and scene graphs, alongside text prompts (20; 22; 23; 24). However, these methods can be impractical, as they often require detailed supplementary inputs that place a burden on users, may involve the costly integration of large language models, and can introduce undesirable biases in generation.

Hence, a training-free method that does not require additional inputs could be ideal for addressing compositional generation challenges. However, no such solution currently exists for managing spatial relationships in T2I models, despite spatial alignment being a well-explored area. Previous studies have primarily used a center-based approach to model spatial relationships, which assesses image-text alignment by focusing on the centers of detected bounding boxes (13; 25; 26; 27). This approach ignores factors such as the overall shape and size of objects, leading to inaccuracies. Additionally, center-based methods rely on discrete spatial inference, which is incompatible with the continuous and differentiable reward functions required for training-free guided generation.

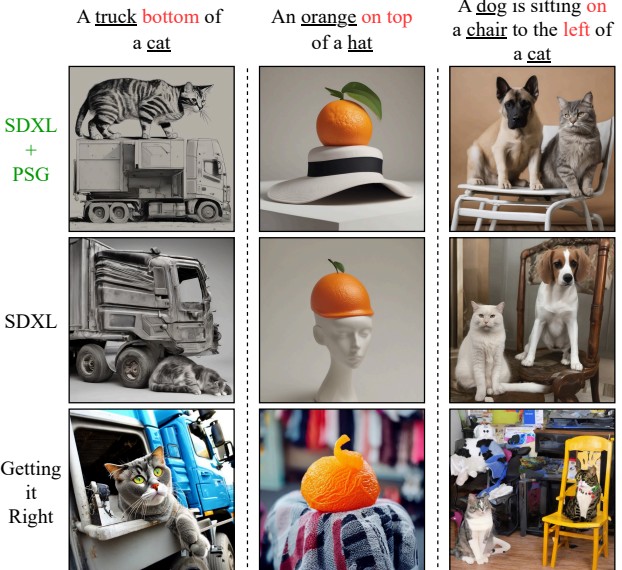

Figure 2: Qualitative comparison with the state-of-the-art SDXL model (1) and the fine-tuning-based model *Getting it Right* (14), on prompts with spatial instructions.

This study introduces a novel probabilistic framework for analyzing spatial relationships between objects using the *Probability of Superiority (PoS)* concept (28), enabling a more human-like interpretation of spatial relationships by accounting for the overall characteristics of objects. Hence, this work presents two main contributions: (1) *PoS-based Evaluation (PSE)*, a novel evaluation metric that reliably assesses 2D and 3D spatial relationship alignment between image and text, and (2) *PoS-based Generation (PSG)*, an inference-time method that enhances T2I models' ability to generate images with specified 2D and 3D spatial relationships by incorporating a PoS-based reward function, without modifying model parameters. Following established paradigms in the field, *PSG* can be implemented in both *gradient-based* and *search-based* forms.

Comprehensive experiments demonstrate that *PSE* aligns more closely with human judgment than commonly used evaluation metrics such as VISOR (13), T2I-CompBench (25), HRS-Benchmark (26), CLIP Score (29), XVLM (30), and Image Reward (31), as measured by Spearman, Kendall, and Pearson correlations. Notably, PSE achieves an improvement of approximately 0.2 in Spearman correlation over VISOR, highlighting that a probabilistic perspective on spatial relationships provides a more reliable assessment than traditional center-based and object detection-based methods.

Additionally, *PSG* outperforms state-of-the-art methods, including fine-tuning-based approaches, across multiple benchmarks and model backbones (fig. 1 and fig. 2). Using the VISOR metric (13), a standard for evaluating spatial relationships in T2I models, *PSG* achieves a significant 32% performance improvement over the leading competitor, while introducing minimal bias, as assessed by quality and diversity evaluations.

Finally, a major challenge in text-image alignment research is the high computational cost of model evaluation, as most compositional metrics require numerous samples for reliable results. This poses difficulties for large-scale models like Stable Diffusion XL (1) and DALL-E 3 (4), underscoring the need for an efficient evaluation metric that reduces sample requirements. Inspired by the approach introduced by (32), we propose *Online PSE (OPSE)*, an online version of the *PSE* metric to reliably detect the superior model using only a limited number of samples.

## 2 Related Work

**Spatial Relationship Generation.** In recent years, various methods have been introduced to address the problem of incorrect spatial relationships in T2I models. The fine-tuning-based approach *Getting it Right* (14) highlights the underrepresentation of spatial relationships in standard vision-language datasets. To address

this, the authors fine-tuned diffusion-based T2I models on a dataset they created by re-captioning six million images from CC-12M (33), Segment Anything (SA) (34), COCO (35), and LAION-Aesthetic (36). Moreover, several fine-tuning methods enhance text-image alignment in T2I models by incorporating additional inputs, such as layouts (37; 38; 39; 40; 41; 42; 43; 44). Since fine-tuning approaches are computationally intensive and prone to overfitting, a line of research has introduced training-free methods that use inference-time approaches to enhance model performance on spatial relationships, often incorporating supplementary inputs (45; 46; 47; 48; 49; 20; 22; 50; 30). These approaches typically adjust intermediate latent embeddings during the denoising process without modifying the model's parameters. To the best of our knowledge, no training-free method currently addresses the spatial relationship problem in T2I models without requiring additional inputs.

**Spatial Relationships Evaluation.** Most evaluation metrics specialized in spatial relationship alignment are center-based, meaning they infer spatial relationships based on the bounding box centers of objects. For example, (13) propose a meric called VISOR which employs the open vocabulary object detector OWL-VIT (51) to assess the spatial relationship alignment by comparing the center points of object bounding boxes identified by the object detector. Moreover, T2I-CompBench (25) and HRS-Benchmark (26), two well-known benchmarks for compositional generation, assess spatial relationships using object centers obtained by UniDet (52). However, center-based approaches have critical limitations as they do not account for the overall characteristics of objects. Additionally, there are also embedding-based and VQA-based evaluation metrics, such as (53) and (54), that address the spatial relationship problem in T2I models.

## 3 Probabilistic Viewpoint to Spatial Relationship

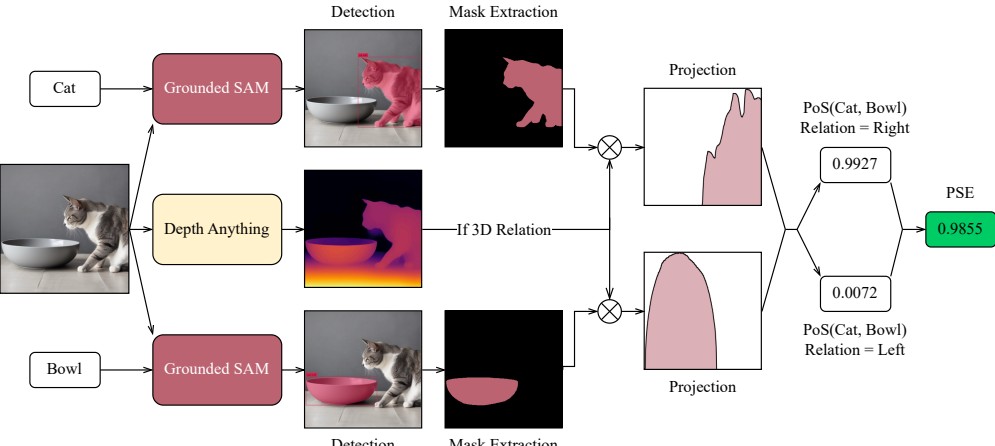

Figure 3: Process of *PSE* metric for the prompt "A cat to the right of a bowl". The depth detection is only applied for 3D relations.

To quantify spatial relationships from a probabilistic perspective, we start with a simple case. Consider two distributions, $A$ and $B$, which, with probability 1, take on single point values $x$ and $y$, respectively. In this scenario, determining whether $A$ is to the right of $B$ reduces to evaluating the condition $x > y$. This can be formally expressed using the indicator function $I(x > y)$, where $I$ returns 1 if the condition holds. This indicator function can be reformulated as shown in eq. (1).

$$I(x, y) = \mathbb{E}_{X \sim A, Y \sim B}[I(X > Y)]$$
$$= P_{X \sim A, Y \sim B}(X > Y) \tag{1}$$

With this intuition, for a more general case, let $A$ and $B$ represent two probability density distributions defined over a one-dimensional space. The concept of distribution $A$ being positioned to the right of distri-

bution $B$ can be quantified by eq. (1) as the likelihood that a randomly selected sample from distribution $A$ is greater than a randomly selected sample from distribution $B$. The formal definition which is known as *Probability of Superiority* ($PoS$) (28) is expressed in eq. (2).

$$PoS(A, B) = P_{X \sim A, Y \sim B}(X > Y)$$
$$= \int_{-\infty}^{\infty} \int_{y}^{\infty} P_A(x) P_B(y) dx dy \tag{2}$$

In a bounded discrete space, given that the range of $A$ is $[s_A, t_A]$ and the range of $B$ is $[s_B, t_B]$, we derive eq. (3).

$$PoS(A, B) = \sum_{j=s_B}^{t_B} \sum_{i=j}^{t_A} P_A(i) P_B(j). \tag{3}$$

Thus, $A$ is considered to be positioned to the right of $B$ when $PoS(A, B) \approx 1$, indicating that the probability of a point in $A$ being to the right of a point in $B$ is nearly 1. Similarly, $A$ is to the left of $B$ if $PoS(B, A) \approx 1$.

To extend beyond one-dimensional distributions, we use normalized projections onto a vector $v$, denoted as $\Pi_v A$ and $\Pi_v B$. These one-dimensional distributions capture spatial relationships along $v$, such as left-right or above-below positioning. Thus, we define $PoS$ on these projections to assess spatial alignment. For example, in 2D, with $\vec{v} = (0, -1)$, $A$ is above $B$ if $PoS(\Pi_v(B), \Pi_v(A)) \approx 1$ and below $B$ if $PoS(\Pi_v(A), \Pi_v(B)) \approx 1$. To simplify notation, we define the *Projected-Pos* for vector $\vec{v}$ as described in eq. (4).

$$PoS_v(A, B) = PoS(\Pi_v(A), \Pi_v(B)) \tag{4}$$

To quantify common directions such as *left*, *top*, or *in front*, we use the standard XYZ axes. The reverse of each relation is modeled by swapping $A$ and $B$ or equivalently using $-\vec{v}$. More complex directions, like *top-right* or *between*, can be captured by combining the *Projected-Pos* of each component.

**Including Distance.** As shown in appendix B.4, eq. (2) can be extended to account for distance, enabling the representation of relations such as *close* and *far*.

Building on the proposed PoS-based framework, we introduce a more reliable evaluation metric for text-image alignment and a training-free method to improve spatial relationship modeling in T2I models.

## 3.1 Evaluating Spatial Relationships Alignment using PoS

Using the concept of *Projected-Pos* (eq. (4)), we introduce a novel evaluation metric, *PSE* (PoS-based Evaluation), to assess the spatial relationship alignment between text and image. Given a pair of text and image, we first extract the object names and their spatial relationships from the text. The image, along with the extracted object names, is then input into Grounded SAM (55), which integrates the Segment Anything Model (SAM) (34) with Grounding DINO (56) to detect and segment the regions related to the objects. For 2D relationships, the *Projected-Pos* is directly applied between the segmentation outputs. For 3D relationships, a monocular depth map detector is first used to estimate the relative depth of objects. The depth values for each segment are then quantized to determine the $\vec{z}$-related values. Thus, for both 2D and 3D cases, the *PSE* evaluation metric for a text-image pair containing objects $A$ and $B$ and their spatial relationship $r$ is defined as eq. (5).

$$PSE(A, B; r) = [PoS_{v_r}(A, B) - PoS_{-v_r}(A, B)]^+ \tag{5}$$

Note that the negative projection vector gives us the *Projected-Pos* for the inverse relation, such as "left" for "right". For example, if object $A$ is entirely to the right of object $B$, meaning all of its points are positioned to the right of $B$'s points, the $PoS_{v_{\text{right}}}(A, B)$ would be close to 1, while $PoS_{v_{\text{left}}}(A, B)$ would approach 0. As a result, the assigned score, $PSE(A, B; Right)$, would be nearly 1, indicating a true alignment. For a visual representation, refer to fig. 3.

The modification in *PSE* to the absolute *PoS* value provides benefits in ambiguous scenarios, where one object is positioned near the middle of another. In such cases, the values for $PoS_{v_{\text{right}}}$ and $PoS_{v_{\text{left}}}$ are nonzero but close, leading to a *PSE* score near zero, which aligns more closely with human evaluation. In contrast, center-based evaluation metrics like VISOR (13) focus solely on the positions of object centers, leading to a rigid scoring of 1 or 0, which favors one object being definitively to the right or left of the other.

*PSE* also offers greater reliability than center-based methods in scenarios where the relative center positions of objects contradict the spatial arrangement that humans intuitively interpret as correct, particularly for objects that can span across the scene, such as trees, as demonstrated in fig. 4.

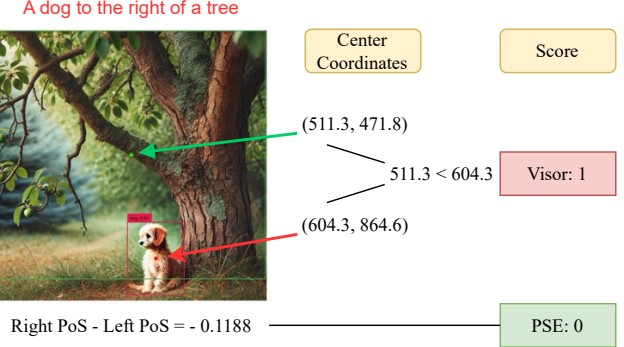

Figure 4: Inconsistency of center-based evaluation with human perception of spatial relationships for the prompt "a dog to the right of a tree". Even though the dog is not positioned to the right of the tree, the tree's horizontal expansion causes the VISOR metric to inaccurately label the relationship between the dog and the tree as true.

Hence, *PSE* provides a human-like measure of confidence in spatial relation evaluation. If all points of one object lie to the right of all points of another, *PSE* confidently assigns a score of $PSE = 1$, indicating that one object is to the right of the other. Moreover, this confidence does not increase further by moving the object farther to the right.

Notably, *PSE* assigns a continuous score between 0 and 1, allowing for the application of a threshold to determine the presence of a specified relationship. Based on our experiments in appendix A.1, we found that a threshold of 0.5 closely aligns with the original *PSE* score, offering an effective binary evaluation method.

## 3.2 Improving Spatial Relationships Alignment using PoS

Building upon the *PSE* metric introduced in section 3.1, we now leverage this metric as a reward function to guide image generation. Specifically, we propose *PSG* (PoS-based Generation), an inference-time alignment enhancement method that steers the generation process to produce images that conform to the spatial configurations described in the input prompt. *PSG* operates without requiring any training or fine-tuning of the underlying generative model. Consistent with established practices in the field, the PoS-based guidance can be implemented through either gradient-based optimization or search-based strategies.

**Gradient-Based *PSG* (Generative Semantic Nursing).** Gradient-based *PSG* can be viewed as a continuous optimization over the latent representations, where the core objective is defined by a PoS-based reward (or loss) function that encourages spatial alignment. This reward can be computed either on the final generated image or on intermediate cross-attention maps extracted during inference; in this work, we primarily focus on the latter. Notably, in T2I models such as U-Net-based architectures (e.g., Stable Diffusion) or DiT-based models, 2D cross-attention maps offer approximate object segmentations before image synthesis is complete. These normalized cross-attention maps can be interpreted as probability distributions over spatial locations, indicating the likelihood that a given object is associated with each region of the image. Given this probabilistic interpretation, the PoS-based formulation (eq. (3)) can be directly

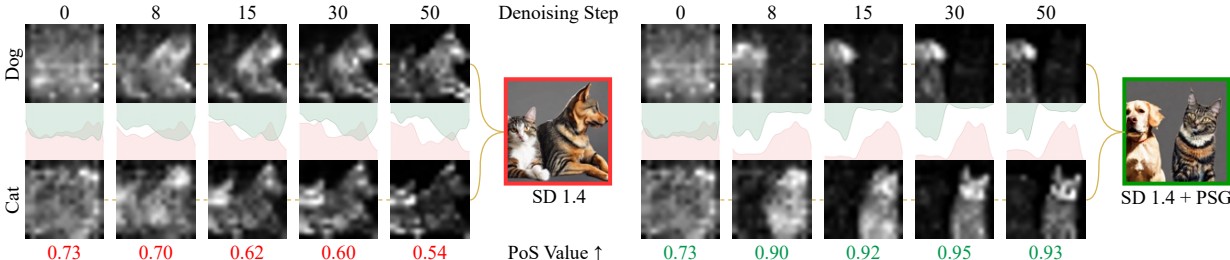

Figure 5: Results of applying gradient-based *PSG* to Stable Diffusion 1.4 on cross-attention maps of dog and cat for the prompt "A dog to the left of a Cat". The distribution of attention of each object is gradually shifted to its correct relative position.

applied to cross-attention maps, enhancing the alignment of 2D spatial relationships during the generation process.

More precisley, consider one of the extracted relations named $r$ (like "being to the left of") between tokens $i$ and $j$ (object associated with token $i$ is to the left of object associated to token $j$). We then use the *Projected-PoS* of the attention maps $A_i$ and $A_j$ on the vector of relation $r$ namely $\vec{v_r}$ as described in the previous subsection. This term now defines a loss $l$ for how much the relation $r$ is established. We employ this loss function to use the concept of generative semantic nursing (6) and update the latent $z_t$ at the denoising step $t$ using eq. (6), where $L$ is equal to the $-PoS^2_{\vec{v_r}}(A_i, A_j)$. The square is due to practical performance.

$$z_t \leftarrow z_t - \alpha_t \cdot \nabla_{z_t} L \tag{6}$$

Notably, fig. 5 depicts the changes in the value of the $PoS_{\vec{v_r}}(A_i, A_j)$ with and without applying eq. (6) to the latent, along with the projected cross-attention maps of entities onto the x-axis. It is apparent that as long as the entities are in an incorrect spatial layout, the value of the proposed PoS-based metric remains low. Combined and more complex form of relationships can be created by merging these losses. For example, by shifting the latent once using the leftward loss and once using the top-ward loss, one can guide towards the top-left relation as presented in fig. 1 . More exemplary instances are provided in appendix B.3.

**Search-Based *PSG* (Inference-time Scaling).** In this context, search-based refers to a discrete search over the initial noise vectors of a diffusion model, serving as an inference-time scaling method. Diffusion models inherently incorporate stochasticity, and prior works (57; 58; 59) have demonstrated that generation quality can be improved by selecting more favorable initial noises. One widely adopted technique for such inference-time enhancement is the *Best-of-N strategy*.

We adopt this approach by using the *PSE* score as a reward function to guide the selection process. Specifically, we generate $N = 32$ samples per prompt and select the image that achieves the highest *PSE* score, as computed by the evaluation pipeline described in section 3.1. While 2D cross-attention maps offer useful spatial cues, they do not inherently capture 3D spatial relationships. In contrast, the final generated images contain sufficient structural and depth information to support 3D-aware evaluation. Consequently, search-based *PSG* is particularly well-suited for enhancing 3D spatial alignment, and it offers a model-agnostic solution that does not require access to the internal architecture or training process of the diffusion model.

## 4 Experiments

We begin by conducting a thorough evaluation of our method's effectiveness in assessing spatial accuracy in images, followed by an analysis of how *PSG* improves alignment with spatial instructions across various benchmarks.

### 4.1 Experiments of PoS-based Evaluation (*PSE*)

We evaluate the performance of our PoS-based evaluation metric, *PSE*, against other established embedding-based and detection-based metrics.

We use images generated by models: SDXL (1), Kandinsky 3 (60), PixArt-$\alpha$ (61), and SDXL+*PSG* to compare the effectiveness of the evaluation metrics on generated images.

Table 1: Correlations on the 2K images of the human evaluation between the evaluators and the human ground truth demonstrates that *PSE* is better correlated with human judgments compared to other metrics. *PSE* (0.5) represents a binary version of *PSE* with a threshold of 0.5.

| Model | Spearman ($\uparrow$) | Kendall ($\uparrow$) | Pearson ($\uparrow$) |
|---|---|---|---|
| VISOR (13) | 0.551 | 0.551 | 0.551 |
| T2I-CompBench (25) | 0.437 | 0.423 | 0.439 |
| HRS-Bench (26) | 0.383 | 0.364 | 0.384 |
| Clip Score (29) | 0.269 | 0.220 | 0.268 |
| XVLM (30) | 0.113 | 0.092 | 0.134 |
| Image Reward (31) | 0.326 | 0.266 | 0.349 |
| *PSE* | 0.726 | 0.642 | **0.778** |
| *PSE* (0.5) | **0.755** | **0.755** | 0.755 |

**Alignment to Human Judgment.** Using the four aforementioned models, we randomly selected 500 prompts from a 5,000-prompt sub-dataset, yielding a total of 2,000 generated images. Three human evaluators, randomly selected from a pool of volunteers, assessed these images. A custom user interface (UI) was developed to present image-prompt pairs in random order, and evaluators were instructed to assign one of the following labels to each pair: (1) object missing, (2) object present but relationally incorrect, or (3) correctly aligned. To mitigate potential bias, we intentionally refrained from providing strict definitions for relational correctness, allowing evaluators to apply their own judgment. The aggregated annotations were subsequently used to compute correlation metrics and analyze the optimism of each scoring method. As presented in table 1, rank correlation analysis indicates that the *PSE* metric achieves the highest alignment with human judgments.

Moreover, *PSE* is specifically designed to handle cases where objects are not well-separated. To evaluate its effectiveness in such scenarios, we selected a subset of images generated by the SDXL model that exhibited higher levels of object occlusion, as identified in a human study. On this challenging subset, *PSE* achieved a substantial improvement in Pearson correlation with human judgments, rising from 0.241 (VISOR metric) to 0.582, demonstrating its superior alignment with human perception compared to center-based metrics, even on difficult samples.

**Reliability Test.** An effective evaluation metric should precisely discern whether a relationship is present or absent in an image. Inaccuracies in the detection of incorrect relationships can lead to overly optimistic assessments. To assess this issue, we evaluated the performance of four metrics -VISOR, T2I-CompBench, HRS-Benchmark, and *PSE* - using Precision, Recall, Accuracy, Specificity, and the F1 Score, with human assessments serving as the ground truth.

In this comparison, we used binary outcomes to score individual image-prompt pairs. VISOR inherently provides binary scores, whereas *PSE* requires thresholding; we classified relationships as correct when the *PSE* score was 0.5 or higher. Similarly, for T2I-CompBench, a threshold of 0.1 was applied to produce binary outputs. As shown in table 2, *PSE* outperforms other metrics in Accuracy, Recall, and F1 Score, indicating it provides more reliable assessments of true relationships. VISOR's low Recall suggests that it tends to overestimate correctness, scoring images too optimistically. In contrast, T2I-CompBench and HRS-Benchmark are overly restrictive, while *PSE* achieves a better balance in this trade-off. The threshold of 0.5 for *PSE* was chosen based on an analysis of the distance between the thresholded and original *PSE*

scores on real datasets introduced by (62). Detailed plots for the hyperparameter selection are provided in Appendix appendix B.1.

Table 2: Comparison of the reliability of binary predictions for spatial relationships. The table shows that VISOR tends to misclassify images with incorrect relationships, while HRS and T2I-CompBench tend to misclassify images with correct relationships. In contrast, *PSE* provides a more balanced evaluation.

| Model | Precision ($\uparrow$) | Recall ($\uparrow$) | Accuracy ($\uparrow$) | Specificity ($\uparrow$) | F1 Score ($\uparrow$) |
|---|---|---|---|---|---|
| VISOR (13) | **86.53** | 62.26 | 76.5 | **90.49** | 72.42 |
| HRS (26) | 40.67 | 72.14 | 73.25 | 73.53 | 52.02 |
| T2I-Compbench (25) | 33.23 | 88.43 | 74.65 | 72.52 | 48.32 |
| *PSE* (0.5) | 78.82 | **88.78** | **88.9** | 88.95 | **83.51** |

**Online PoS-based Evaluation (*OPSE*).** Both VISOR and *PSE* metrics rely on datasets with a large number of generated images to provide robust assessments of generative models, using 30k and 5k prompts, respectively. However, when selecting the model with the highest *PSE* value among multiple candidates, acquiring the necessary number of prompts can be costly, particularly due to the API expenses of large-scale models. To reduce this cost, rather than performing batch sampling—which risks allocating resources to suboptimal models—we propose leveraging the Multi-Armed Bandit approach (63), as suggested by (32; 64), for model selection. As shown in fig. 6, employing the UCB algorithm (65; 66) effectively converges to sampling from the optimal model based on the ranking in table 3. Further details on the algorithm are provided in Appendix appendix A.6.

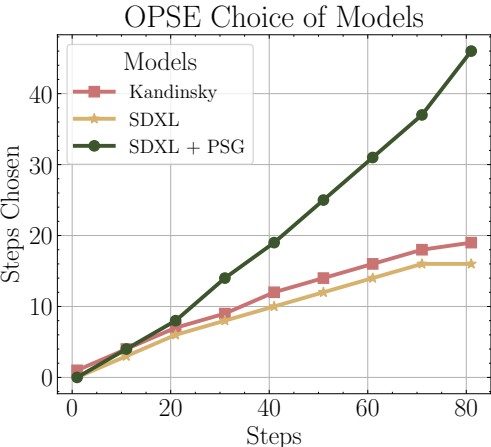

Figure 6: Number of times each model is chosen by the *OPSE* algorithm for 100 rounds. The models are SDXL, Kandinsky, and SDXL + *PSG*. We observe that *OPSE* can accurately find the best model and sample from it.

## 4.2 Experiments of PoS-based Generation (*PSG*)

**Experimental Setup.** We evaluated *PSG* on two backbone models: Stable Diffusion 1.4 (2), a lightweight model with limited capacity, and Stable Diffusion XL (1), a more advanced and computationally powerful model. For Stable Diffusion 1.4, *PSG* was applied during the first 25 steps of the denoising process, using an initial step factor of 20. In contrast, for Stable Diffusion XL, the loss function was applied over the first 10 steps, with a larger initial step factor of 1000. The impact of these hyperparameter choices is further analyzed in appendix B.1. We used an NVIDIA A100 GPU for the experiments. Three well-known compositional generation benchmarks were adopted for comparisons.

Table 3: Comparison of *PSE* 2D and 3D scores of models used for our human experiment. (This is not a performance validation for *PSG* as it uses the same measure).

| Model | $PSE_{2D}$ (%) ↑ | $PSE_{3D}$ (%) ↑ |
|---|---|---|
| PixArt-$\alpha$ | 17.20 | 27.21 |
| Kandinsky 3 | 26.07 | 28.52 |
| SDXL | 20.34 | 28.60 |
| SDXL+*PSG* | 65.89 | 78.10 |

Table 4: Quantitative comparison of our gradient-based *PSG*, compared to baselines using VISOR metrics on VISOR dataset.

| Model | OA (%) ↑ | $VISOR_{uncond}$ (%) ↑ | $VISOR_{cond}$ (%) ↑ | $VISOR_1$ (%) ↑ | $VISOR_2$ (%) ↑ | $VISOR_3$ (%) ↑ | $VISOR_4$ (%) ↑ |
|---|---|---|---|---|---|---|---|
| GLIDE (67) | 3.4 | 1.9 | 55.96 | 6.36 | 1.06 | 0.16 | 0.02 |
| GLIDE + CDM (19) | 9.96 | 6.33 | 63.47 | 19.7 | 4.6 | 0.88 | 0.12 |
| DALLE-mini (68) | 27.24 | 16.21 | 59.53 | 39 | 17.5 | 6.42 | 1.94 |
| CogView2 (69) | 18.86 | 12.31 | 65.26 | 33.78 | 11.6 | 3.4 | 0.44 |
| DALLE-v2 (70) | 63.59 | 37.42 | 58.85 | 72.78 | 46.16 | 23.08 | 7.66 |
| SD 1.4 + CDM | 23.24 | 15.02 | 64.63 | 38.98 | 14.62 | 5.22 | 1.26 |
| SD 1.4 (2) | 29.6 | 18.84 | 63.65 | 46.16 | 20.12 | 7.26 | 1.82 |
| SD 2.1 (2) | 48.66 | 30.9 | 63.5 | 65.2 | 36.5 | 17.1 | 4.8 |
| Getting it Right (14) | 60.55 | 44.05 | 72.75 | 71.14 | 52.66 | 33.92 | 18.48 |
| SD XL (1) | 65.74 | 45.01 | 68.46 | 78.52 | 55.38 | 32.98 | 13.16 |
| SD 1.4 + *PSG* (1) | 35.29 | 30.0 | 85.0 | 60.48 | 34.74 | 18.32 | 6.46 |
| SD 1.4 + *PSG* (2) | 44.92 | 39.98 | 89.0 | 71.58 | 47.72 | 28.56 | 12.06 |
| SDXL + *PSG* (1) | 76.8 | 74.8 | **97.4** | 93.52 | 84.72 | 70.88 | 50.08 |
| SDXL + *PSG* (2) | **80.02** | **77.35** | 96.6 | **94.98** | **87.02** | **74.36** | **53.04** |

**VISOR Benchmark.** VISOR (13) is a widely used benchmark framework for assessing spatial relationships in T2I models. It comprises using 30,000 prompts based on the objects of COCO (35) to test object positioning in four key directions: left, right, top, and bottom. 5,000 prompts with four random seeds to where used compare *PSG* with re-evaluated baseline models. The detailed explanation of the benchmark is postponed to appendix A.4. Notably, table 4 shows that *PSG* not only enhances spatial relationship accuracy but also improves Object Accuracy (OA) by mitigating object overlap issues. Based on $VISOR_{uncond}$ metric, our method achieves a 30% increase over prior methods and models. Moreover, on VISORcond, it attains an impressive 97%, highlighting consistent spatial alignment. Additionally, it improves $VISOR_{1/2/3/4}$ by 20%, surpassing baseline methods by achieving a higher probability of spatial accuracy across different random seeds. We further tested a prompt-simplified *PSG* variant ("(2) versions in table 4") that replaces explicit spatial terms with "and". This modification boosted OA by 9% for SD 1.4 and 3% for SDXL. Notably, *PSG* on SD 1.4 outperformed SD 2.1, despite SD 2.1's overall advantage on VISOR metrics.

**T2I-CompBench and HRS-Benchmark.** We further evaluated *PSG* using T2I-CompBench (25) and HRS-Benchmark (26), focusing on their spatial relationship assessment. As shown in table 5, *PSG* improves SDXL's spatial alignment on T2I-CompBench by 15% over standard SDXL and 8% over DALLE-3. Results for SD 1.4 are deferred to appendix B.2. Similarly, table 6 shows that *PSG* enhances spatial accuracy on HRS-Benchmark, outperforming layout-guided generative models. T2I-CompBench evaluates 3D spatial relations using detection scores and IoU constraints in addition to comparing the depth center of objects. However, as we focus on evaluating the generation of correct relative 3D positions, in addition to T2I-CompBench, we used specified experiments for this matter. We used Grounded SAM and Depth Anything as more recent detectors to extract depth center of objects. We then evaluate the spatial accuracy by comparing the center points. In another experiment, we also included an IoU constraint similar to T2I-CompBench. We used a threshold of 0.5 for the object detection and $N = 32$ (section 3.2) for applying *PSG*. As shown in table 7, our method not only improves SDXL's performance on T2I-CompBench but also achieves gains in depth-focused 3D spatial assessments.

Table 5: Quantitative comparison of baselines using T2I-CompBench spatial benchmark for gradient-based *PSG* on SDXL backbone.

| Model | T2I Spatial Score (%) ↑ |
|---|---|
| GORS (25) | 0.1815 |
| Getting it Right (14) | 0.2133 |
| PixArt-$\alpha$ (61) | 0.2064 |
| Kandinsky v2.2 (71) | 0.1912 |
| SD-XL (1) | 0.2133 |
| DALLE-3 (4) | 0.2865 |
| SDXL + *PSG* | **0.3601** |

Table 6: Quantitative comparison of gradient-based *PSG*, compared to baselines with Stable Diffusion backbone using HRS spatial benchmark (scores of other models are adopted from GrounDiT (72)). This benchmark consists of prompts that establish relationships among multiple objects within each prompt.

| Model | Spatial (%) ↑ |
|---|---|
| Stable Diffusion (2) | 8.48 |
| PixArt-$\alpha$ (61) | 17.86 |
| Layout-Guidance (20) | 16.47 |
| Attention-Refocusing (49) | 24.45 |
| BoxDiff (48) | 16.31 |
| R&B (50) | 30.14 |
| SDXL + *PSG* (ours) | **34.33** |

Table 7: Quantitative comparison of search-based *PSG* method with the SDXL baseline in generating 3D spatial relationships.

| Model | T2I-CB 3D (%) ↑ | Depth w/ IoU (0.2) (%) ↑ | Depth w/o IoU (%) ↑ |
|---|---|---|---|
| SDXL | 35.66 | 28.46 | 43.10 |
| SDXL + *PSG* | **37.13** | **33.15** | **63.67** |

**Embedding-based Alignment Analysis.** Beyond object detector-based metrics, which primarily assess object presence in generated images, we also evaluated text-image alignment using the embedding-based CLIP Score. This metric measures the similarity between a generated image and its corresponding prompt within the CLIP model's embedding space (73). Applying the *PSG* to SDXL on the COCO dataset captions (35) improves the CLIP Score from 0.313 to 0.316, underscoring *PSG*'s effectiveness in enhancing alignment.

**Quality and Diversity Results.** Controlling generation in diffusion models by modifying latent embeddings during denoising often degrades image quality (74). To evaluate these side effects, we selected 20,000 image-caption pairs from MS-COCO 2017, ensuring that each pair contained at least one distinct spatial relationship between two objects, identified via bounding box annotations. The captions were then augmented to explicitly specify object positioning. We generated images using SDXL alone and SDXL with *PSG*, comparing their quality and diversity metrics against COCO images as the reference (table 8). For feature extraction, we used DINOv2-ViT-L/14 (75). Additionally, FID scores were reported using Inception-V3. Results indicate a slight quality reduction in FID (76) and CMMD (77) scores, aligning with *PSG*'s emphasis on spatial alignment rather than pure image quality. Given the known concerns associated with the FID metric (1), we also evaluated Precision and Recall (78) metrics, both of which demonstrated performance levels close to the baseline. Additionally, the RKE metric (79), a reference-free measure of diversity,

indicated a higher mode count with *PSG* compared to SDXL alone, confirming that *PSG* enhances mode diversity.

Table 8: Quality and diversity comparison of our proposed evaluation method, *PSG*, with baseline SDXL on COCO dataset.

| Model | | Quality | | | Diversity | |
|---|---|---|---|---|---|---|
| | CMMD ($\downarrow$) | FID ($\downarrow$) | Precision ($\uparrow$) | Recall ($\uparrow$) | RKE($\uparrow$) |
| SDXL | **0.77** | **221.07/20.23** | **0.85** | 0.59 | 401.63 |
| SDXL + *PSG* | 0.79 | 238.05/21.52 | 0.83 | **0.60** | **412.73** |

**Diversity of Bounding Boxes.** Layout-guided generative models, while capable of placing objects in specific positions, rely on manually defined layouts, introducing significant bias into the generated images. To examine this phenomenon, we analyzed the template of RealCompo (80) for bounding box generation in appendix B.5. The experiments demonstrate that methods relying on LLM-generated layouts exhibit limited spatial diversity in object position and size. In contrast, our proposed approach (*PSG*) produces significantly more diverse and natural layouts in the final images, better reflecting real-world variability.

**Generalization to Other Backbones.** We also applied our gradient-based *PSG* method on top of transformer architecture and search-based *PSG*, i.e., Best-of-N *PSG*, on various state-of-the-art models which demonstrated that the improvements are not limited to the choice of the model. The results are postponed to appendix B.2.

## 5 Conclusion

This work introduces a novel probabilistic framework for modeling spatial relationships between objects, inspired by the concept of *Probability of Superiority (PoS)*. The main contributions are as follows: (1) *PSE* (PoS-based Evaluation), which provides smoother and more human-aligned assessments of 2D and 3D spatial relations than conventional center-based metrics by accounting for the holistic characteristics of objects; (2) *PSG* (PoS-based Generation), a training-free method that utilizes a PoS-based reward function, applied either through a continuous gradient-based approach or a discrete search-based strategy, resulting in superior spatial relation alignment in both 2D and 3D compared to state-of-the-art methods; and (3) *OPSE*, an online variant of *PSE*, which enables efficient and reliable evaluation of T2I models using a minimal number of samples. The limitations of both contributions are discussed in appendix A.5 and appendix B.8.

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

# A  *PSE*

## A.1  *PSE* Threshold Hyper-parameter

What'sUp (62) introduces several dataset of images labeled according to the spatial relationships between objects depicted in the images. To compare *PSE* with other evaluators using metrics such as Precision, Recall, Accuracy, Specificity, and F1 Score, we need to determine an appropriate threshold. For this purpose, we analyze the performance of both the default *PSE* and its thresholded version on two real-world datasets from What'sUp. The first dataset is What'sUp Section A, which consists of 408 images with 2D spatial relationship captions. The second dataset is COCO-Spatial, containing 2,687 images with spatial relationship captions. As shown in appendix A.2, in the first experiment, thresholds of 0.4 and 0.5 yield results closest to the non-thresholded version of *PSE*, while in the second experiment, thresholds of 0.5 and 0.6 provide the closest results. Based on these observations, we select 0.5 as the threshold for the thresholded version of *PSE*.

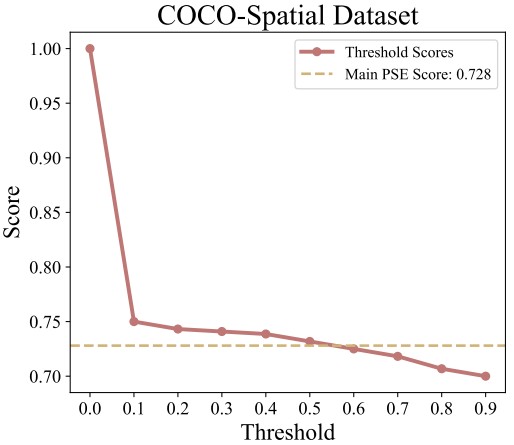

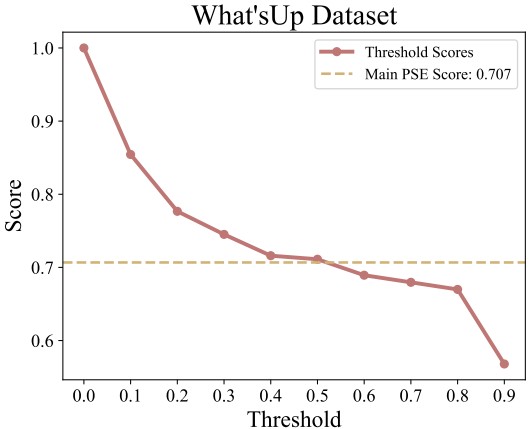

(a) The results of thresholded version of *PSE* for different threshold values on the COCO-Spatial dataset.

(b) The results of thresholded version of *PSE* for different threshold values on the What'sUp dataset.

Figure 7: Thresholded version of *PSE* for different threshold values on the COCO-Spatial and What'sUp datasets.

## A.2  Visual Clarification of Ineffectiveness of Using Centers

fig. 4 illustrates that relying on the center of bounding boxes, as done by other detection-based evaluators, can lead to approving incorrect relationships. Even using the mask centers of each object does not resolve this issue. For example, the x-coordinates of the mask centers for the dog and tree are 612.0 and 599.8, respectively, which again results in the dog being incorrectly identified as to the right of the tree.

Another challenge arises when an object is neither to the right nor the left from a human perspective. For instance, in fig. 8, we observe that the chair is neither to the right nor the left of the TV. The centers of the bounding boxes and masks for the TV and chair are detailed in table 9. The x-coordinate of the chair's bounding box center is less than that of the TV's bounding box center. Consequently, metrics based on center-based directional relationships, such as those used by VISOR and HRS-Benchmark, would incorrectly classify the chair as being to the left of the TV. Similarly, segmentation-based mask centers fail to address this issue, as the chair's mask center is also positioned to the left of the TV's center.

In contrast, *PSE* evaluates this scenario more accurately. The *PSE* values for both left and right relationships are notably low for the image, indicating that it does not consider either relationship a valid description of the scene.

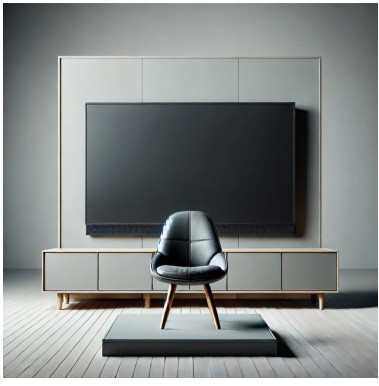

Figure 8: A failure case of VISOR metric. VISOR incorrectly determines the chair to the left of the TV, while the chair is neither to the right nor left of the TV.

Table 9: Centers of bounding boxes and masks for the TV and chair. The *PSE* Left score is also reported.

| Object | Box Center X | Box Center Y | Mask Center X | Mask Center Y | *PSE* LEFT |
|--------|--------------|--------------|---------------|---------------|------------|
| Chair | 507.68 | 729.54 | 507.86 | 693.94 | 0.01 |
| TV | 512.71 | 454.53 | 510.91 | 446.20 | 0 |

## A.3 Robustness of Detector

Grounded-SAM was selected for its reported superior performance over commonly used models (55). However, to make sure that our score is robust enough to use on segmentations extracted from models like Grounded-SAM, we used 100 randomly selected VISOR prompts and generated images using SDXL backbone. We (1) apply three mask corruptions and observe minimal impact on *PSE* (table 10); (2) perform *PSG* (inference time scaling) with $N = 4$ and *PSE* across multiple detectors, noting consistent scores (table 11). These results show that first, *PSE* is robust to common corruptions of the mask and also the type of detector used for mask extraction.

Table 10: Dropout removes x% of the mask, Jitter randomly moves the mask x pixels, and Morph erodes and dilates with a 3x3 kernel.

| Corruption | Dropout | | | | Jitter | | | | Morph (Opening) | | | |
|------------|---------|-------|-------|-------|-------|-------|-------|-------|-------|-------|-------|-------|
| | 10% | 30% | 50% | 80% | 5px | 20px | 50px | 100px | 1 it | 2 it | 5 it | 10 it |
| *PSE* Score | 29.40 | 29.39 | 29.40 | 29.37 | 29.75 | 29.83 | 29.77 | 31.32 | 29.40 | 29.41 | 29.50 | 30.44 |

Table 11: *PSE* and *PSG* best-of-N ($N = 4$) with combination of various detectors. * indicates a non-*PSE*, over-optimistic metric.

| Model | SDXL | +Grounded-SAM (*PSG*) | +ClipSeg (*PSG*) | +Mask2Former(*PSG*) |
|-------|------|----------------------|------------------|---------------------|
| Grounded-SAM (*PSE*) | 29.39 | 53.71 | 48.86 | 45.07 |
| CLIPSeg (*PSE*) | 30.68 | 44.78 | 52.04 | 42.31 |
| Mask2Former (*PSE*) | 26.43 | 41.23 | 40.45 | 46.79 |
| *Gounding-Dino (Box Center) | 43.00 | 77.50 | 63.00 | 62.75 |

In addition, we note that the baseline spatial detection metric (VISOR) uses OWL-ViT. As our goal is to detect objects in generated images, we compared how OWL-ViT performs in comparison to Grounded-SAM on generated images on 500 SDXL generated images. We used the prompts from our human evaluation experiment. We compared how many times each detector missed an object which was labeled as presented by the human evaluators. Grounded-SAM showed a 5.8% detection miss rate, which was significantly lower

than OWL-ViT at 11%. Therefore, Grounded-SAM is a reasonable choice for a detector. Even though, by advancements in object segmentation, our method can be applied on top of any other detector too.

## A.4   Center-based Evaluation Metrics

Center-based evaluation metrics assess the alignment of spatial relationships between the generated image and the input prompt by analyzing the bounding box centers of objects identified by an object detector. For instance, if the center of one object's bounding box is positioned to the right of another, these metrics infer that the first object is located to the right of the second. However, unlike our proposed PoS-based evaluation metric (*PSE*), center-based metrics fail to account for critical factors such as object shape and size. This limitation often leads to inaccuracies, particularly in complex scenes with intricate spatial relationships.

Through this work, we evaluated the performance of the *PSE* in comparison to three center-based evaluation metrics as follows.

**VISOR metrics.**   The VISOR benchmark introduces six evaluation metrics to assess spatial relationship alignment in generated images. Among these, the $\text{VISOR}_{\text{uncond}}$ metric evaluates spatial accuracy by determining whether both objects are accurately generated and positioned as specified in the prompt. In contrast, the $\text{VISOR}_{\text{cond}}$ metric measures spatial accuracy conditional on the correct generation of both entities. Another key metric, $\text{VISOR}_{\text{n}}$, calculates the success rate of generating $n$ out of 4 images (where $n$ ranges from 1 to 4) that adhere to the spatial relationships described in the prompt. To score spatial relationship alignment, the VISOR benchmark utilizes the open vocabulary object detector OWL-ViT (81), which identifies objects in the image and compares the center points of their bounding boxes to assess compliance with the spatial relationships specified in the textual input.

**T2I-CompBench and HRS-Bench Metrics.**   T2I-CompBench and HRS-Benchmark are two prominent benchmarks designed to evaluate compositional generation tasks in text-to-image (T2I) models. In order to assess the spatial relationship alignment between the generated image and the input prompt, both benchmarks utilize center-based evaluation metrics by employing UniDet (52) as the object detector to extract the bounding boxes of objects. Spatial relationship accuracy is then determined by comparing the relative positions of the bounding box or depth centers, providing a measure of how well the generated images adhere to the specified spatial relationships in the input prompts.

## A.5   Limitations

Our proposed PoS-based evaluation metric (*PSE*) offers a smoother and more nuanced assessment of spatial relationships compared to center-based approaches by adopting a probabilistic perspective that accounts for the overall shape and size of objects. However, similar to center-based methods, *PSE* is inherently influenced by the performance of the depth estimation or segmentation model it relies on. Specifically, *PSE* determines an object's mask in the generated image using segmentation outputs provided by models such as SAM. Consequently, any limitations or inaccuracies in these tools can propagate through the evaluation process, potentially affecting the reliability of the overall assessment.

## A.6   Online PoS-based Evaluation Details

As discussed in the main paper, one drawback of using VISOR or our sub-dataset for *PSE* evaluation is the need to generate thousands of images for each model to conduct a reliable comparison. For users seeking to identify the best-performing model among several candidates, this can be prohibitively expensive due to the current API costs associated with image generation. To address this issue, we suggest a technique for evaluating models in an online manner, which avoids the need for generating large batches of images. This approach reduces the computational burden of sampling from suboptimal models.

We adopt a framework similar to that introduced in (32). Let the set of generative T2I models be denoted as $\mathcal{G} := [G]$, where each generator $g \in \mathcal{G}$ produces images distributed according to $p_g$ over a dataset $D$. The evaluation process is carried out over $T$ steps, where, at each step, the algorithm selects a generator $g_t$ from

$\mathcal{G}$ and collects a set of samples $x_t \sim p_g$ generated by the chosen model. The goal is to design an algorithm that effectively balances exploration and exploitation to identify the optimal model (i.e., the one with the highest *PSE* score).

A notable feature of *PSE* is its linearity—scores for individual samples are independent and additive. This property enables the use of established methods for optimizing the selection process. Specifically, the score of a generator $g$ can be expressed as $\mathbb{E}_{x \sim p_g}[PSE(x)]$, allowing the application of tools like Hoeffding's inequality (82) to compute optimistic bounds. In this context, $PSE(x)$ is the *PSE* score for individual samples, and the mean of the IID samples serves as an empirical estimate of the generator's true score. Using these properties, we employ the Upper Confidence Bound (UCB) algorithm, a well-established approach in the multi-armed bandit framework, to dynamically identify the best-performing model. The IID property of the samples arises from the use of random seeds and prompts for image generation.

During each step of sampling, the model with the highest upper confidence bound for its empirical *PSE* score is selected. Let $n_t(g_i)$ denote the number of times model $i$ has been chosen up to step $t$. The upper confidence bound for the $i$-th model at step $t$ is defined as:

$$PSE_{UCB}(g_i; t) = \frac{\sum_{x \in D_t(g_i)} PSE(x)}{n_t(g_i)} + \alpha \sqrt{\frac{\ln(t)}{n_t(g_i)}} \tag{7}$$

where $\alpha$ controls the probability of failure (selecting a suboptimal model with a lower true score). Based on empirical observations, we set $\alpha = 2$ in our experiment. At each step $t$, the generative model with the highest $PSE_{UCB}(g_i; t)$ value is selected. This strategy minimizes the risk of excessive sampling from suboptimal models, ensuring a more efficient and accurate evaluation process.

# B  *PSG*

## B.1  Hyper-parameters

Our Generative Semantic Nursing method involves two key hyperparameters: the number of denoising steps during which noise optimization is applied and the scaling factor of the loss function. To determine optimal values for these hyperparameters and ensure fairness in our main experiments, we employed a set of ten objects not included in the COCO dataset. From this set, 100 pairs were randomly selected and assigned random spatial relationships. Images were then generated using the *PSG* method, implemented on both the Stable Diffusion 1.4 and Stable Diffusion XL backbones.

For *PSG* applied to Stable Diffusion XL, we experimented with scaling factors ranging from 100 to 2500 in increments of 900 and optimization step counts ranging from 4 to 22 in increments of 6. The VISOR metric was used to evaluate spatial accuracy. Our results indicate that increasing the scale factor and step count generally improves adherence to the specified spatial relationships. However, beyond certain thresholds, these increases lead to a degradation in the quality of the generated images. For instance, a low scale factor provides insufficient guidance, resulting in outputs resembling SD-XL's default behavior, as shown in fig. 9. Through experimentation, we determined that a configuration of (10 steps, 1000 scale factor) offered an optimal balance, achieving better spatial relationship adherence without compromising image quality. Illustrative examples of the effects of high and low step counts and scale factors are provided in fig. 9, fig. 11, and fig. 12.

For *PSG* applied to Stable Diffusion 1.4, optimal performance was achieved with approximately 20 steps and a considerably smaller scale factor. After experimenting with scale factors and step counts ranging from 10 to 25, we identified the best configuration as 20 steps and a scale factor of 25. In contrast, applying *PSG* to Stable Diffusion XL required significantly fewer steps (at most 10) keeping the time overhead of *PSG* relatively low. On an NVIDIA A100 GPU, applying *PSG* during the first 10 steps resulted in less than a 20% increase in processing time of SD1.4. Further reducing the number of steps can decrease this overhead even more, maintaining the method's efficiency. Figure 10 illustrates the scalability of our methods based on PoS when additional computation is available. It further shows that gradient-based latent optimization is generally more effective, while search-based methods provide a way to exploit unbounded computation for continued performance gains.

Furthermore, we apply Gradient-Based *PSG* using the $32 \times 32$ cross-attention maps, based on the analysis in table 12 (on 500 sample prompts), which demonstrates their superiority.

Table 12: Comparison of different U-Net layers for Gradient-Based *PSG* shows the superiority of the $32 \times 32$ cross-attention maps of SDXL.

| Metric | Without *PSG* | 32×32 Maps *PSG* | 64×64 Maps *PSG* |
|--------|---------------|------------------|------------------|
| VISOR  | 44.0          | **74.4**         | 48.2             |
| PSE    | 21.0          | **62.0**         | 32.0             |

## B.2  Further Quantitative Results

The overall spatial score, as evaluated using the HRS-Benchmark (26), is presented in table 6. Additionally, the score, segmented by prompt difficulty, is shown in table 13. Scores for other models are taken from the supplementary materials of the HRS-Benchmark. The T2I-CompBench 2D spatial score for SD 1.4 is also reported in table 14.

**Generalization to Other Backbones.**  For inference time scaling *PSG*, we additionally used 100 randomly selected prompts from the VISOR prompts and up to 1.6K images to analyze the behavior when aplied on different models. Table table 16 shows that models with various architectures can use our method

Table 13: Quantitative assessment of our proposed generative semantic nursing method *PSG* on top of SDXL, based on the HRS-Spatial benchmark split on the prompt difficulty.

| Model | Easy | Medium | Hard |
|-------|------|--------|------|
| SDV1 | 21.75 | 0 | 0 |
| SDV2 | 1.19 | 0 | 0 |
| Glide | 2.49 | 0 | 0 |
| CogView 2 | 8.88 | 0 | 0 |
| DALL-E V2 | 28.34 | 0 | 0 |
| Paella | 8.78 | 0 | 0 |
| minDALL-E | 4.29 | 0 | 0 |
| DALL-EMini | 15.17 | 0 | 0 |
| *PSG* | **68.56** | **28.44** | **5.98** |

to improve their alignment to spatial constraint defined in the prompt. Furthermore, table 16 also suggests that by using more compute (increasing $N$), we can consistently achieve better performance.

**Generative Semantic Nursing On Transformer Architecture.** We applied *PSG* on the average cross-attention maps of PixArt-$\alpha$ as a transformer-based model. We used a scale 10 and applied guidance

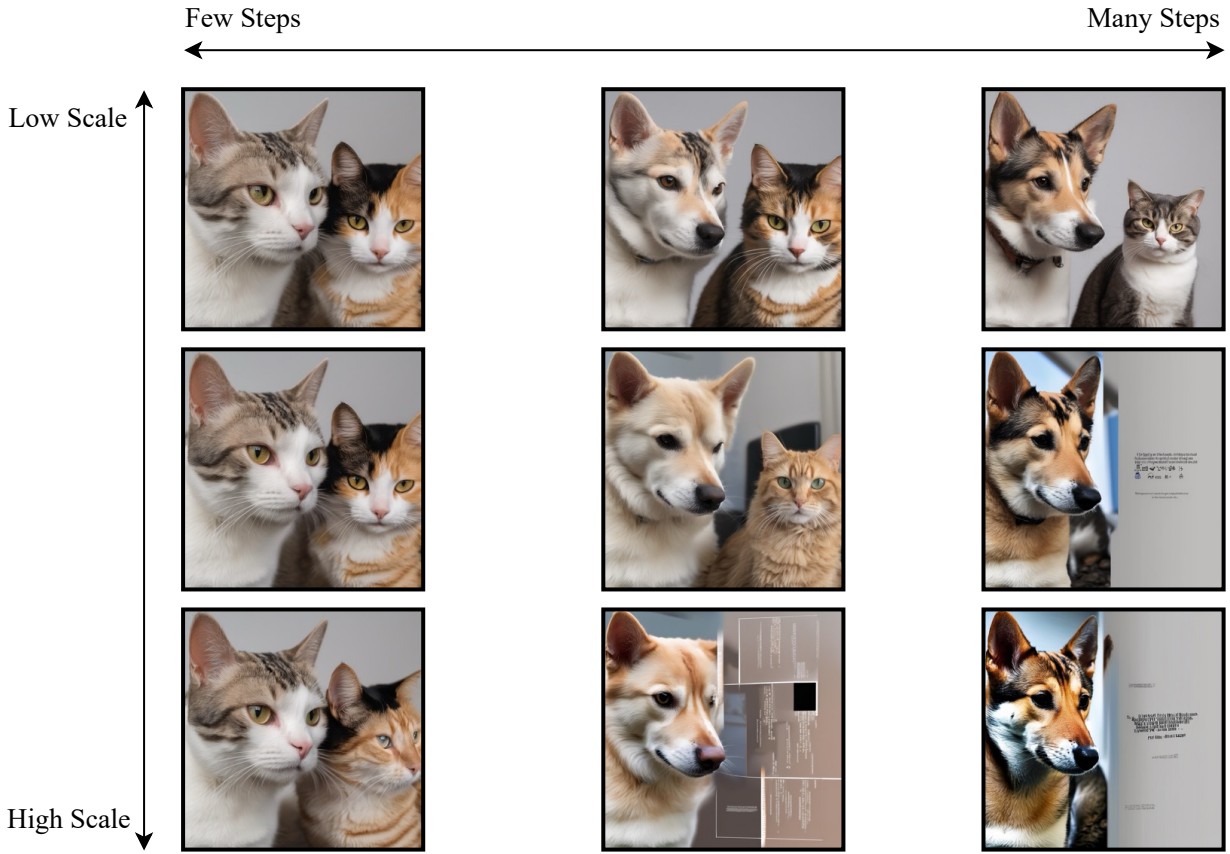

Figure 9: Qualitative analysis of the effect of the scale factor of *PSG*'s loss function and the number of denoising steps it is applied, for the prompt "a dog to the left of a cat".

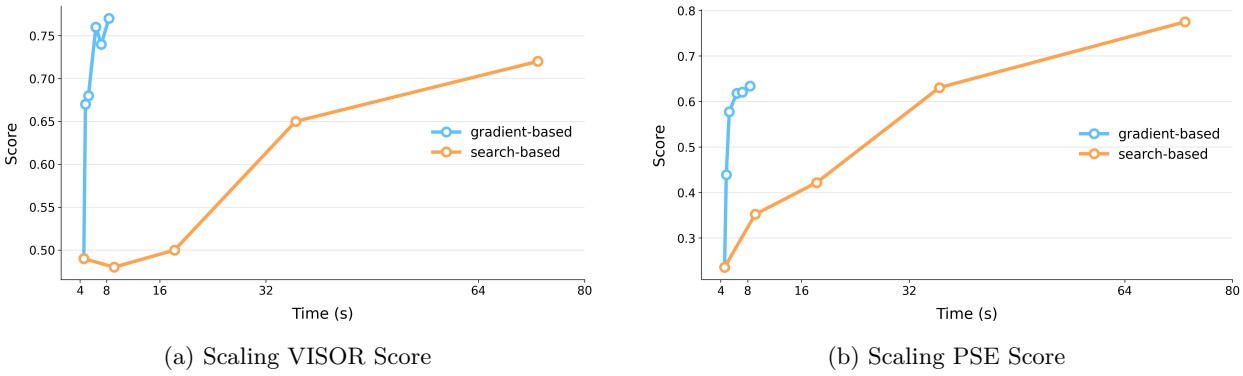

(a) Scaling VISOR Score

(b) Scaling PSE Score

Figure 10: Scaling behavior of search-based and gradient-based PSG by scaling the number of images in Best-of-N and increasing the number of time-steps during which the latent optimization is applied.

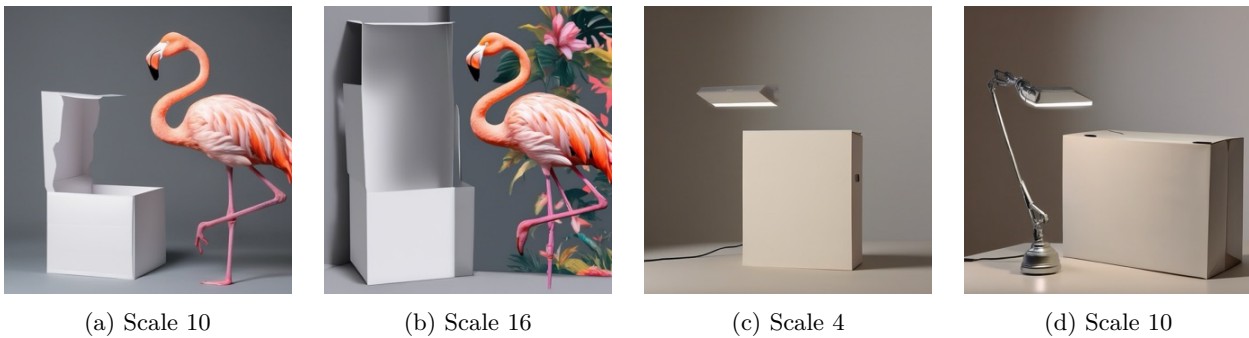

(a) Scale 10         (b) Scale 16         (c) Scale 4         (d) Scale 10

Figure 11: The effect of the scale factor in the *PSG* method. As the scale increases, the generated images can become more distorted. Using a medium scale instead of a very small one can prevent incomplete images.

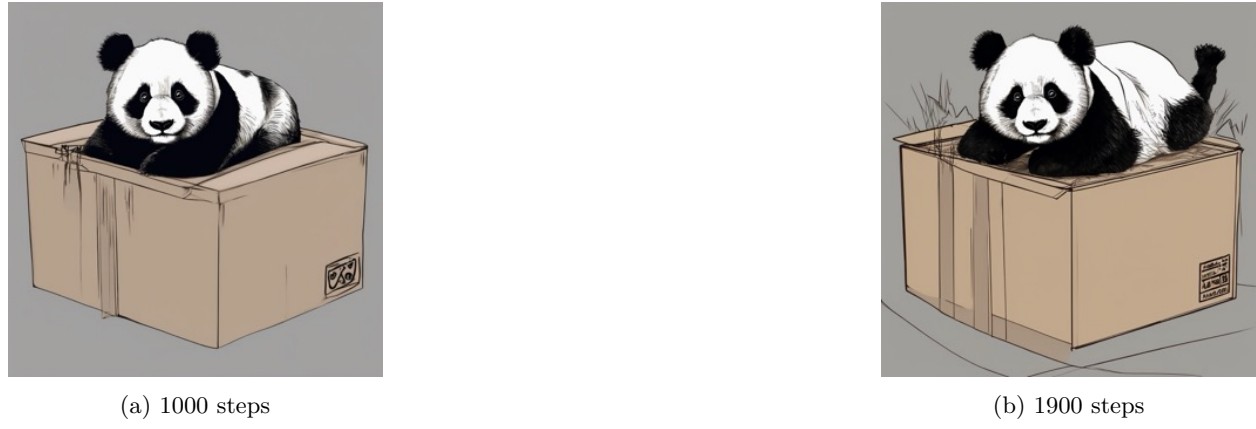

(a) 1000 steps                                  (b) 1900 steps

Figure 12: The effect of optimization step counts in the *PSG* method. A higher number of steps distorts the images more, resulting in lower quality.

for 20 steps. Using the same prompts as the previous experiment, *PSE* improved from 28.50 to 57.87, and center-direction accuracy increased from 40% to 68%. This indicates that our method is not limited by the U-Net architecture and can also be applied on transformer-based generative models.

**Additional Complex Prompts.** Beyond table 6, which includes prompts with multiple relationships, we evaluate *PSG* on two sets of 100 prompts. The first set contains three objects with two spatial relationships

Table 14: Quantitative comparison of baselines using T2I-CompBench spatial benchmark for *PSG* on SD 1.4 backbone.

| Model | T2I Spatial Score(%) ↑ |
|---|---|
| SD 1.4 (2) | 0.1246 |
| SD 2 (2) | 0.1342 |
| Composable v2 (19) | 0.0800 |
| Structured v2 (12) | 0.1386 |
| Attn-Exct v2 (6) | 0.1455 |
| DALLE-2 (70) | 0.1283 |
| SD 1.4 + *PSG* | 0.2349 |

Table 15: Quantitative comparison of SDXL and SDXL+*PSG* on *PSE* and VISOR metrics (average score of the relationships) for 100 prompts with three and four objects.

| # Objects | Model | PSE | VISOR |
|---|---|---|---|
| 3 | SDXL | 30.8 | 57.0 |
|   | SDXL+*PSG* | **63.8** | **72.5** |
| 4 | SDXL | 31.5 | 54.0 |
|   | SDXL+*PSG* | **68.4** | **58.0** |

Table 16: *PSE*/(center position accuracy) of *PSG* inference time scaling. This table demonstrates the effectiveness of our method for different models. Increasing $N$, shows that by using more compute we can get consistently better alignment. We observe that applying *PSG*, even with small $N$, has a noticeable effect on the accuracy of position of objects.

| Model | w/o *PSG* ↑ | *PSG* N=2 ↑ | *PSG* N=4 ↑ | *PSG* N=8 ↑ | *PSG* N=16 ↑ |
|---|---|---|---|---|---|
| FLUX | 45.52 / 60 | 56.85 / 73 | 72.05 / 82 | 83.60 / 91 | 93.59 / 96 |
| SANA1.5 | 72.62 / 82 | 80.31 / 89 | 87.81 / 95 | 90.58 / 97 | 93.71 / 99 |
| SD3 | 47.41 / 52 | 64.29 / 70 | 75.36 / 86 | 82.58 / 91 | 88.13 / 94 |
| SDXL | 29.39 / 43 | 40.07 / 64 | 55.56 / 76 | 67.14 / 86 | 78.23 / 93 |
| SD2.1 | 16.07 / 32 | 29.25 / 50 | 43.70 / 75 | 60.94 / 89 | 77.52 / 95 |

(e.g. "a balloon on the bottom of a chicken and on the left of a lamp"), following the VISOR multi-object template. The second set contains four objects with three relationships (e.g. "a cup on the right of a dog and a bicycle on the left of a man"). Table 15 demonstrates the effectiveness of *PSG* on these more complex prompts.

**Search-Based VISOR.** Since the search-based variant of *PSG* can also be applied with alternative center-based verifiers such as VISOR, we conducted a direct comparison between the two. Specifically, we used Best-of-16 SDXL-generated images on a subset of 100 prompts from table 1. In table 17, we report both the *PSE* and VISOR scores for these images. Because using the same verifier for both generation and evaluation risks reward hacking (i.e., each method achieves its highest score under its own verifier), we additionally performed human evaluation. Annotators assigned a score of 1 if all objects were present and their spatial relationships correct, and 0 otherwise. Table 17 shows that *PSG* consistently outperforms VISOR as a search-based method. This result aligns with our earlier finding that *PSE* correlates more strongly with human judgments.

Table 17: Quantitative comparison of search-based methods on *VISOR*, *PSE*, and Human evaluation.

| Method | VISOR ↑ | PSE ↑ | Human ↑ |
|---|---|---|---|
| Search-Based-VISOR | **92.0** | 33.8 | 22.0 |
| Search-Based-PSG | 72.0 | **77.5** | **61.0** |

### B.3 Further Qualitative Results

Figures 15 and 16 provide a visualization of the effectiveness of the *PSG* method compared to the baseline model (Stable Diffusion XL) and the fine-tuning method ("Getting it Right"). The similarity in the overall appearance of the images for each prompt between the baseline model and the *PSG*-applied version can be attributed to the use of the same seed for generating images across all models. fig. 17 further illustrates the accuracy of spatial relationships demonstrating that *PSG* maintains correct spatial relationships while adhering to the stylistic features specified in the text prompt.

Additionally, fig. 18 presents a qualitative assessment of *PSG*'s performance on combined spatial relationships. The results demonstrate that *PSG* accurately infers spatial relationships, even in complex scenarios involving combinations of primary directions. For prompts that focus on providing 3D instructions, please refer to fig. 19.

### B.4 Incorporating Distance

Distance can be incorporated into PoS to model proximity (e.g., close or surrounding) and separation (e.g., far). eq. (2) can be extended by introducing a threshold $c$, which defines a distance constraint, resulting in the following PoS variation:

$$PoS^d(A, B; c) = P_{X \sim A, Y \sim B}(X > Y + c) \tag{8}$$

where $c$ controls the degree of separation, either as a fixed constant or a function of object sizes. Summing projected $PoS^d$ across the four principal directions yields:

$$
\begin{aligned}
PoS^{\text{distance}}(A, B; c) = &PoS^d_{right}(A, B; c) + \\
&PoS^d_{left}(A, B; c) + \\
&PoS^d_{top}(A, B; c) + \\
&PoS^d_{bottom}(A, B; c)
\end{aligned}
\tag{9}
$$

This formulation rewards or penalizes deviations beyond $c$ pixels (by inverting the loss sign), effectively incorporating distance modeling. Notably, summing over all four directions dissipates directional effects, making it roughly equivalent to the IoU loss used for attention binding (7). To evaluate this modification, we conducted experiments on 100 multi-object prompts with fixed random seeds and $c = 5$ (corresponding to an attention-map resolution of 16 by 16). The far loss increased the average $L_2$ norm distance between object mask centers from 102.5 pixels to 130.8, while the close loss reduced it to 90.7 pixels, demonstrating that this approach effectively improves distance-based 2D relations.

### B.5 Diversity of Bounding Boxes.

The dependency of Layout-guided generative models, not only undermines the model's autonomy but also fails to address the core goal of understanding and correctly applying relational positions directly from the prompt without external input. As the text is informative enough about the spatial relationships. In contrast, our model uses the prompt itself to guide the diffusion process, ensuring spatial relationships are applied naturally during generation.

We explored an automated alternative where Large Language Models (LLMs) suggest layouts based on prompts (47; 80). However, in an experiment generating 50 images from 4 prompt sets (e.g., "an object to

the right of another"), LLM-generated bounding boxes exhibited low diversity, with bounding box centers and sizes frequently clustering. This lack of variation reduces the natural diversity expected in real-world scenarios. In contrast, our approach with SDXL + *PSG*, driven purely by the prompt, produces significantly more diverse bounding boxes compared to RealCompo's template (80) for bounding box generation using GPT-4o (83), as confirmed by table 18.

Table 18: Variance of center and size of boxes for SDXL + *PSG* (Generative Semantic Nursing) and RealCompo (80) shows that our method's effect has better diversity of size and position of objects compared to using LLM guided bounding box for objects.

| | Model | Center A ($\uparrow$) | Center B ($\uparrow$) | Size A ($\uparrow$) | Size B ($\uparrow$) |
|---|---|---|---|---|---|
| 1 | SDXL + *PSG* | **0.127** | **0.095** | **0.109** | **0.063** |
| | RealCompo | 0.036 | 0.07 | 0.041 | 0.025 |
| 2 | SDXL + *PSG* | **0.092** | **0.111** | **0.169** | **0.291** |
| | RealCompo | 0.036 | 0.042 | 0.037 | 0.059 |
| 3 | SDXL + *PSG* | **0.088** | **0.056** | **0.125** | 0.161 |
| | RealCompo | 0.05 | 0.051 | 0.058 | **0.174** |
| 4 | SDXL + *PSG* | **0.081** | **0.159** | **0.129** | **0.281** |
| | RealCompo | 0.043 | 0.038 | 0.068 | 0.09 |

## B.6 Ablation Studies

**Super-category Pairs Split Score.** In fig. 14, we visualize the performance of our method, *PSG*, alongside the baseline backbones, based on the super-category pairs of objects. The super-categories are derived from the COCO dataset. It is evident that our models consistently improve the VISOR score. The poor performance on the "person-person" pair is due to the fact that, in the VISOR dataset, there is only one "person-relation-person" pair. Additionally, we observe that the performance of our method is also dependent on the initial performance of the backbone. When the backbone performs better initially, our method approaches optimal accuracy.

**Order Bias.** fig. 13 suggests that the accuracy of backbones for the first and second objects is not consistent. In our experiment, both Stable Diffusion 1.4 and XL demonstrate higher accuracy for the first object. While our method primarily focuses on enhancing spatial relationships, we observe that it consistently improves the overall accuracy of object generation. Notably, in Stable Diffusion XL, which already exhibits good accuracy in generation, our method reduces the number of missing objects for each generated image.

**Relation Split Scores.** We have presented the Object Accuracy and VISOR score for each relation type in Table table 19. Our method has improved both object accuracy and VISOR score across all relationships. Additionally, it is worth noting that the performance of our model on Left and Right relationships is somewhat better than on Top and Bottom relationships. We hypothesize that this difference is due to the relative rarity of Top and Bottom relationships in the VISOR dataset. This effect of rarity is further illustrated in fig. 14, where we observe lower performance on pairs of outdoor objects when compared to indoor objects and appliances, as these pairs are less likely to appear together in real image datasets.

## B.7 Using LLMs for Spatial Relationship Extraction

We suggest using the following prompt, which is specifically engineered to output the object and relationship pairs for our loss function. Additionally, a code is provided to identify the tokens associated with the objects using Stable Diffusion's tokenizer.

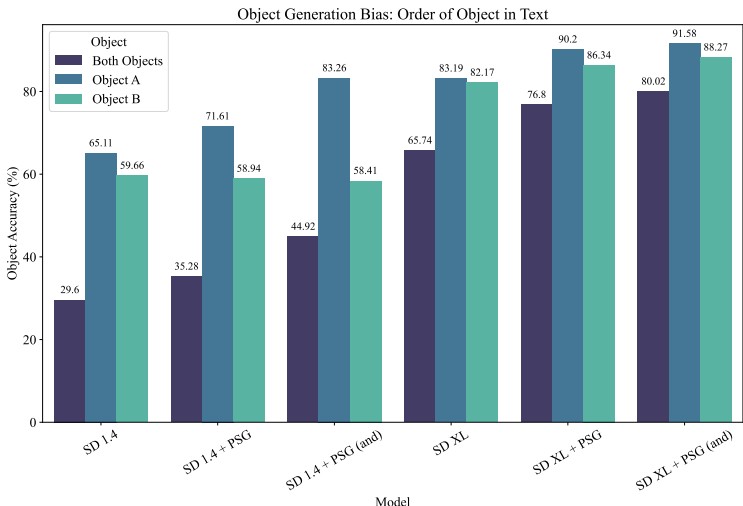

Figure 13: Order bias in generated images with *PSG* variants applied on SD 1.4 and SDXL backbones. In all experiments, *Object A* refers to the first object and *Object B* refers to the second object.

---

**Example LLM Instruction**

Analyze the provided input text to extract spatial relationships between objects, then output these relationships in a structured format. Follow these steps precisely:

1. **Identify Spatial Relationships**: Locate any spatial relationships, either explicitly stated or implicitly conveyed. Possible relationships are "left," "right," "top," "bottom," "behind," or "in front." Note relationships that may be implied by context as well.
*Example:*
Input: "A horse is running to the left of a car."
Relationship: **left**
2. **Identify Related Objects**: For each spatial relationship, determine the two objects involved.
*Example:*
Object 1: **horse**
Object 2: **car**
3. **Format the Output**: Present each object pair and their spatial relationship in the format `(object1, object2, relationship)`, with each pair on a new line.
*Example:*
Input: "The man is reading a book while sitting. A dog is sitting to the right of his chair."
Output:
`(man, chair, top)`
`(dog, chair, right)`
**Task**: Apply this process to the following text and provide only the final output in the specified format. Use step-by-step reasoning but keep the format of the final response.
*Input Text:*

---

## B.8 Limitations

Our proposed training-free method provides a computationally efficient, on-the-fly solution for addressing spatial relationship challenges in T2I models. However, it is important to acknowledge certain limitations that can be explored and addressed in future research.

**Limitations of the training dataset of T2I models.** Spatial relationship-related examples are often underrepresented in large-scale multi-modal datasets. Additionally, these datasets may contain various spatial relationship-based biases. Such issues are likely key contributors to the poor performance of T2I

Table 19: Comparison of Visor and Object Accuracy scores of our models and the backbones split by relationship type shows that our method consistently improve the accuracy of prompts with each direction.

| Models | VISOR (%) (↑) | | | | Object Accuracy (%) (↑) | | | |
|---|---|---|---|---|---|---|---|---|
| | Left | Right | Top | Bottom | Left | Right | Top | Bottom |
| SD 1.4 | 18.71 | 18.23 | 22.01 | 16.26 | 29.32 | 28.9 | 33.31 | 26.68 |
| SD 1.4 + $PSG$ | 30.64 | 31.4 | 34.36 | 23.54 | 34.25 | 34.86 | 41.08 | 30.71 |
| SD 1.4 + $PSG$ (and) | 41.52 | 39.7 | 40.31 | 38.4 | 44.89 | 44.06 | 45.1 | 45.59 |
| SD XL | 43.14 | 45.53 | 46.74 | 44.55 | 62.56 | 64.7 | 67.43 | 68.06 |
| SD XL + $PSG$ | 77.29 | 77.18 | 72.75 | 72.25 | 78.29 | 78.15 | 75.5 | 75.44 |
| SD XL + $PSG$ (and) | 82.18 | 80.83 | 74.16 | 72.67 | 83.46 | 82.09 | 77.42 | 77.41 |

models on spatial relationship benchmarks. Consequently, training-free generation methods that guide the diffusion model during inference may not offer a comprehensive solution to the spatial relationship problem, particularly for complex text prompts involving numerous entities. In such cases, extensive fine-tuning of T2I models on carefully curated datasets tailored to specific domains may be necessary to achieve optimal performance. However, this fine-tuning process typically demands substantial computational resources and time, highlighting the need for future research to strike a balance between training-free methods and fine-tuning-based approaches.

**Limitations of the CLIP text encoder.**  The limited capability of the CLIP text encoder to accurately understand and represent objects and their spatial relationships may significantly affect the performance of T2I models in generating spatially coherent images. As most T2I models rely on the CLIP text encoder as the backbone for processing textual inputs, any deficiencies in the encoder's ability to capture and encode spatial relationships can propagate through the denoiser, resulting in suboptimal outputs or outright failures in the generated images. Surprisingly, much of the existing research has overlooked this potential root cause, often attributing these failures to other factors without adequately considering the inherent limitations of the CLIP text encoder. To address this gap, our future work will focus on enhancing the text encoding capabilities of T2I models, leveraging the PoS-based perspective proposed in this study.

**Memory Usage and Computation Time.**  The amount of available information for aspects such as spatial relationships depends on the generative model used. For instance, standard 2D text-to-image diffusion models, such as Stable Diffusion, do not provide information about the third axis—depth—in an image. We proposed a method for generative semantic nursing, which introduces minimal overhead to the model. However, in cases where the model does not inherently provide 3D spatial relationships, we employ initial noise search using PoS. This approach requires segmentation and/or depth estimation models, introducing some computational and memory overhead. Our experiments show that the time overhead is negligible, but as models scale, higher GPU memory will be required. However, the significant improvement in compositional generation achieved by $PSG$ justifies this trade-off. Importantly, $PSG$'s reliance on external estimators is not intrinsic; for instance, in 2D spatial relationships, it can directly leverage the architecture of T2I models, eliminating the need for additional estimators.

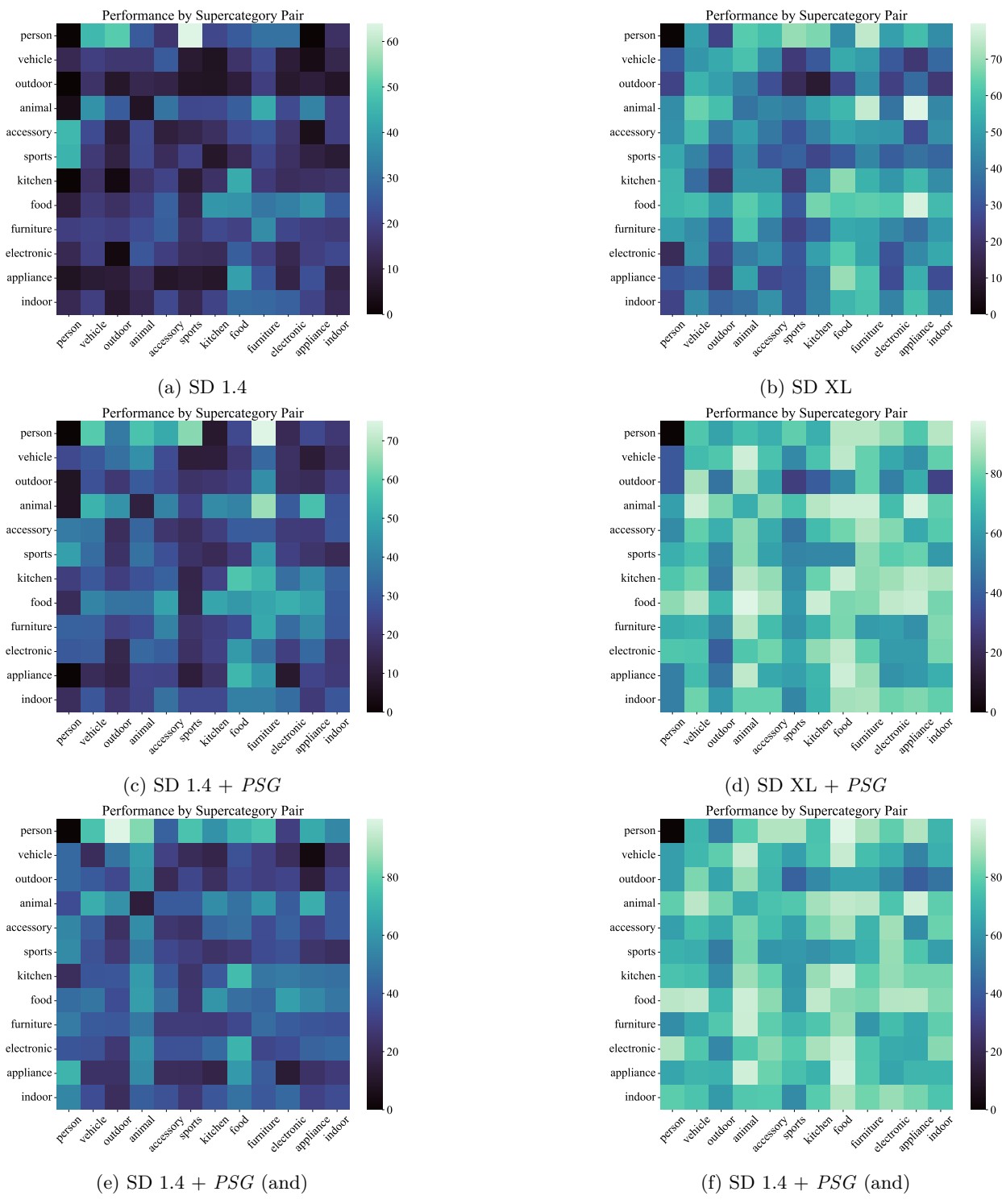

Figure 14: VISOR unconditional scores of our models and baselines split by super-category pairs. We observe that our models consistently achieve better VISOR unconditional scores.

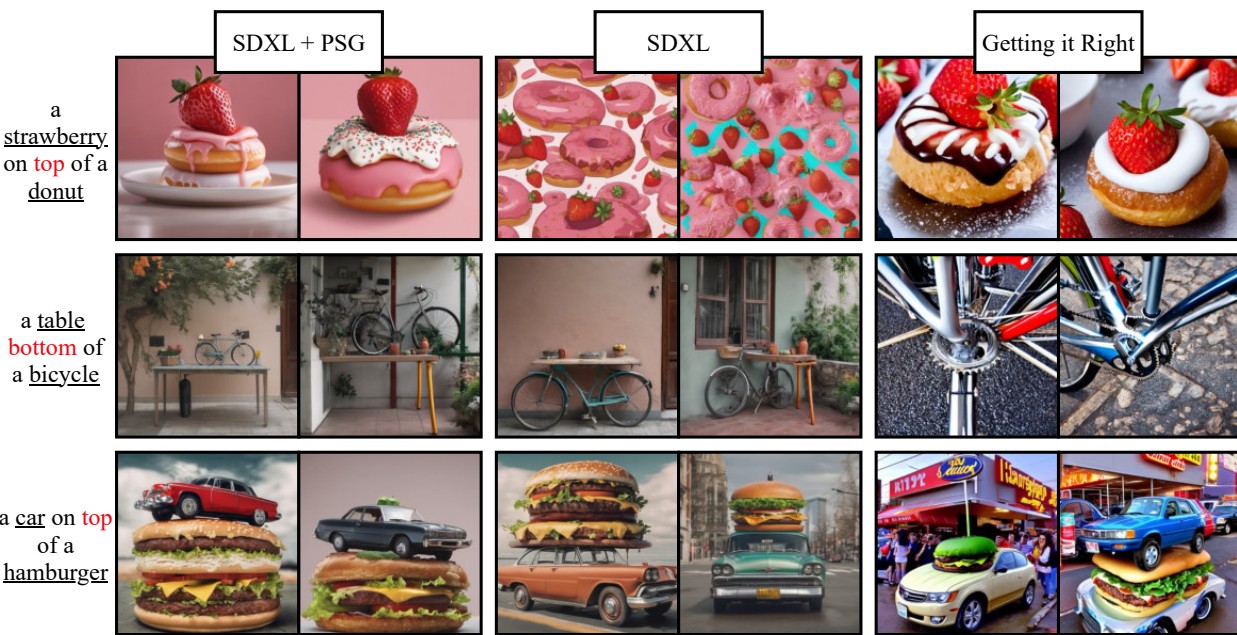

Figure 15: Qualitative comparison of our proposed method, *PSG*, versus Stable Diffusion XL and Getting it Right on spatial relationship-related prompts.

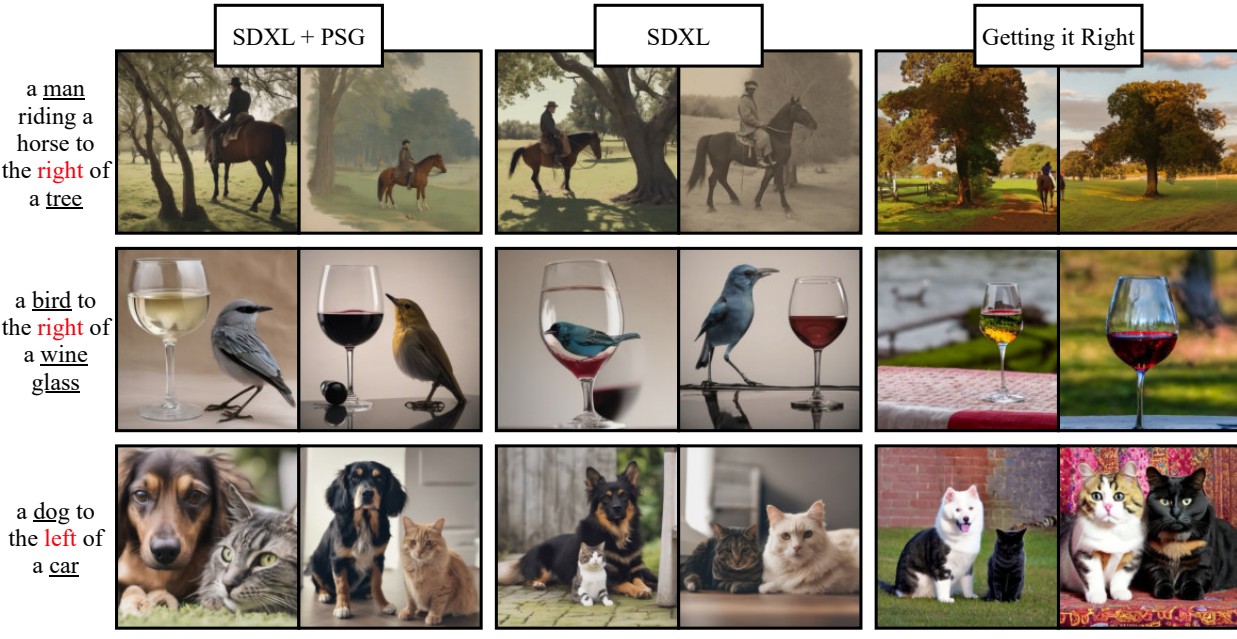

Figure 16: Qualitative comparison of our proposed method, *PSG*, versus Stable Diffusion XL and Getting it Right on spatial relationship-related prompts.

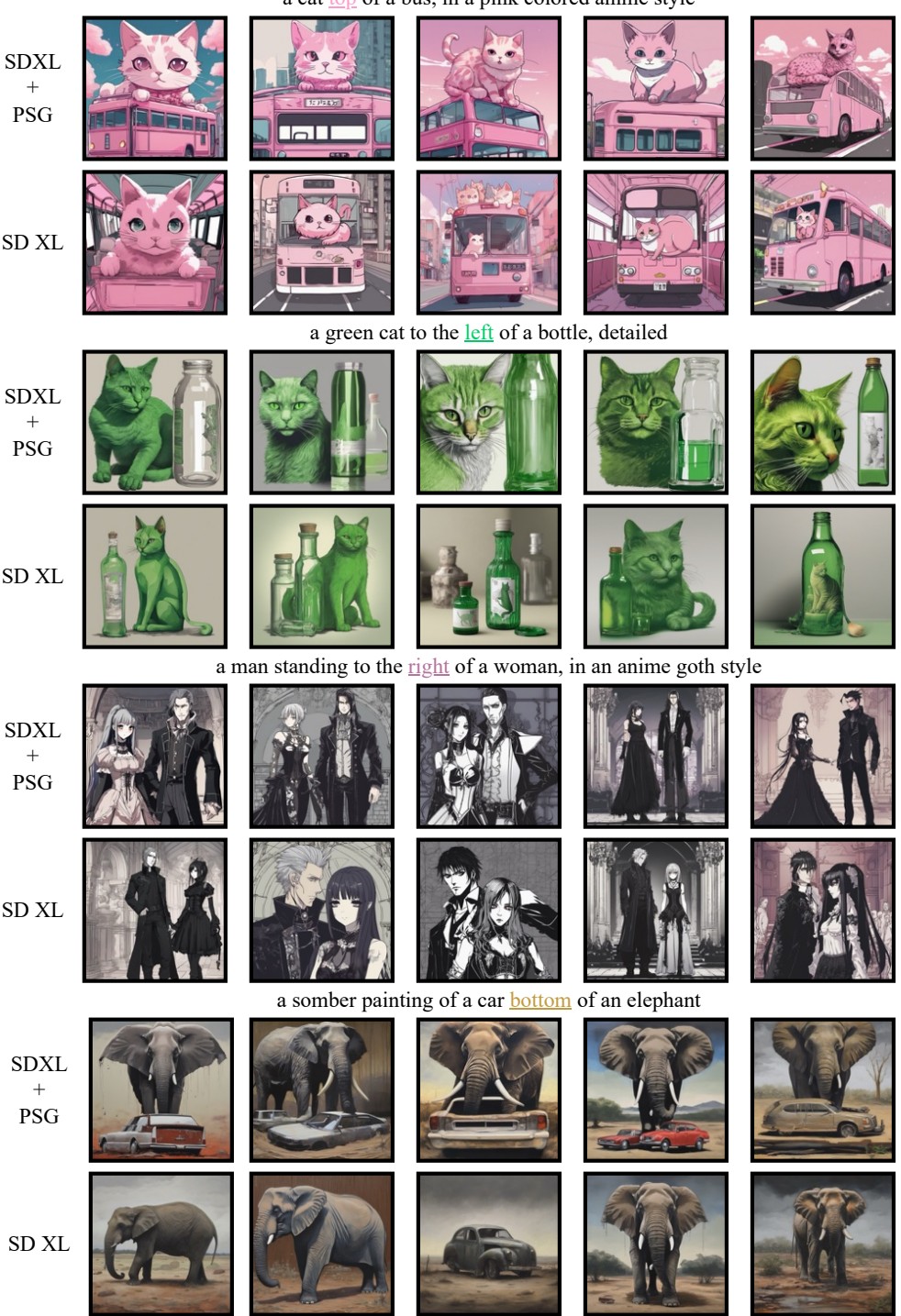

Figure 17: Qualitative comparison of our proposed method, *PSG*, versus Stable Diffusion XL on artistic prompts.

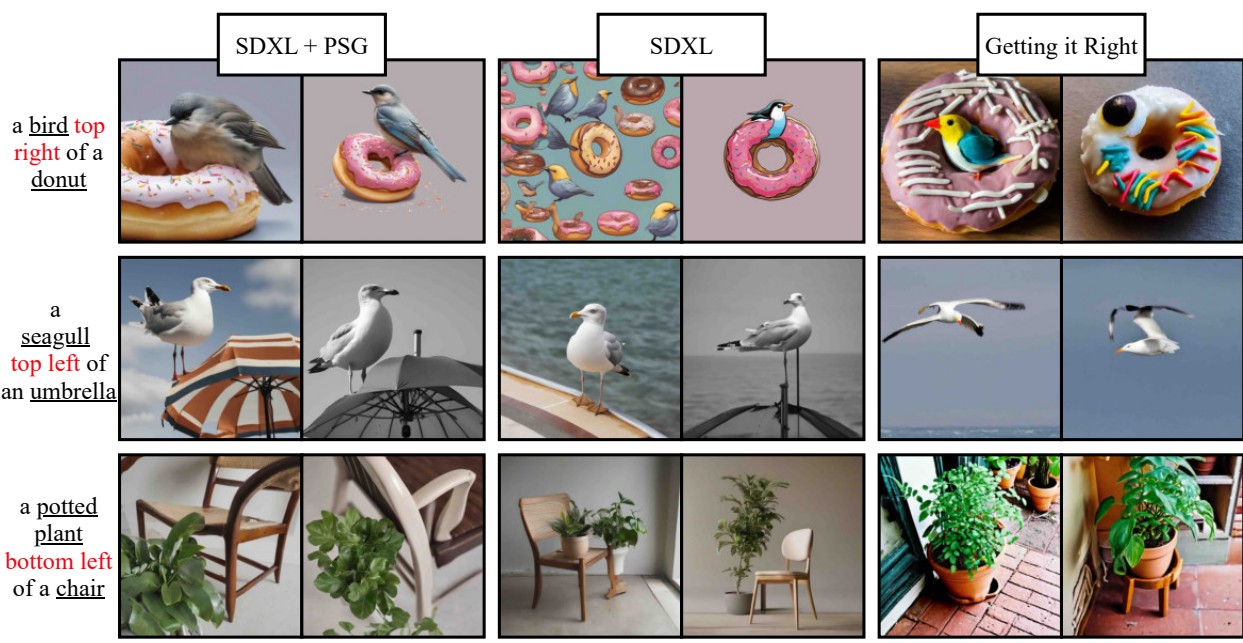

Figure 18: Qualitative comparison of our proposed method, *PSG*, versus Stable Diffusion XL and Getting it Right on prompts with combined spatial relationships,

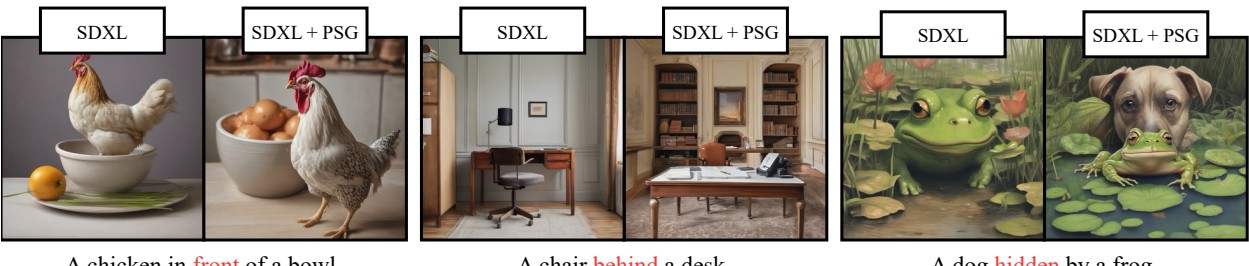

Figure 19: Qualitative comparison of our proposed method, *PSG*, versus Stable Diffusion XL on 3D instructions. Similar to T2I-CompBench, "hidden" is treated as equivalent to "behind".

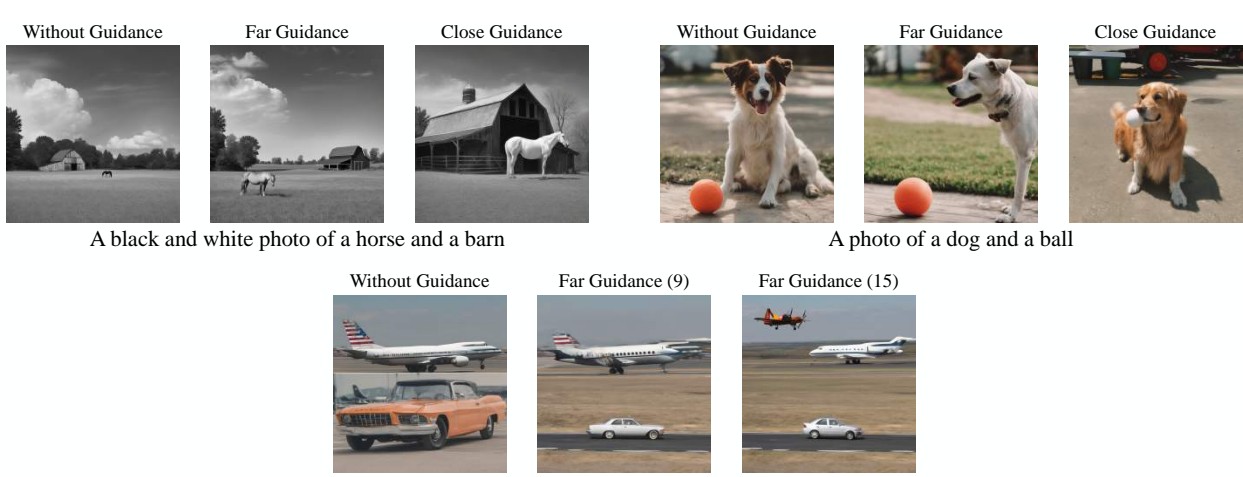

Figure 20: Illustration of close vs. far guidance using PoS with the distance loss described in appendix B.4. The hyperparameter $c$ (e.g., 9 or 15 in the bottom figure) specifies the threshold for what is considered far.

