# OpenReview forum: "Why Settle for Mid: A Probabilistic Viewpoint to Spatial Relationship Alignment in Text-to-image Models"
_TMLR — Accepted by TMLR_

### Review · Reviewer_4mMA · 2025-07-28

**Summary Of Contributions:**

The paper proposes a framework based on Probability of Superiority (PoS) to evaluate the spatial relationships of objects as specified in text prompts in diffusion models. The proposed method utilizes the cross-attention maps generated by the diffusion model for each object in the text prompt. Using these cross-attention maps, a PoS metric is computed to determine the 2-D spatial location of the probability densities relative to each other. The PoS metric is also extended to 3-D object generation by incorporating depth maps generated using Depth Anything. PoS is further used as a training free technique to improve generation performance of pretrained diffusion models.

**Audience:**

Yes

**Claims And Evidence:**

Yes

**Requested Changes:**

1. The paper needs more in-depth explanation on how the probabilities and projections are computed.
2. The paper needs to show an ablation study that determines the best layer to be used for computing the cross-attention map for PoS.
3. The papers needs to include the performance and robustness of the model with respect to the number of objects in the text prompt. Ideally, show the performance of the method with 2, 3, 4, .. etc objects in the text prompt.

**Strengths And Weaknesses:**

**Strengths**

1. A PoS based technique to determine the spatial orientation of objects is proposed. The motivation and formulation of the framework is sound and meaningful.
2. The PoS based metric outperforms traditional metrics such as VISOR and CLIP score.

**Weaknesses**

1. It is unclear from the text how the probabilities and the projections are computed for computing the evaluation metric in Section 4.1. Which layer of the U-Net diffusion model is used to compute the cross-attention map? Does the selection of the layer affect the performance?
2. How generalizable is the proposed method when generating multiple objects with multiple relationships among themselves? Also, how robust are the metrics when evaluating spatial relationships with multiple objects?
3. Can the formulation be extended to include relations such as "covered" in the 2-D case and "behind" in the 3-D case?
4. Can this formulation be used for editing images? For example, can it be used for swapping different objects in the image? The paper should discuss more in-depth implications about their proposed method for future applications.
5. Can the PoS based generation be used as an additional loss when training a diffusion model from scratch? Currently, it seems it can only be applied on pre-trained diffusion models.

---

> ### Author Response · Authors · 2025-09-26
> **Response to reviewer 4mMa [Part 1]**
>
> We thank the reviewer for the positive and encouraging feedback. We are pleased that the motivation and formulation of our PoS-based framework for determining spatial orientation were found to be sound and meaningful. We also appreciate the acknowledgment that our proposed evaluation metric outperforms traditional baselines such as VISOR. Below we address the reviewer’s remaining concerns in detail.
>
> 1- **Projection computation and U-Net layer ablation**
>
> As described in Section 3, the probabilities are derived either from normalized cross-attention maps or from segmentation masks, both of which yield a 2D spatial distribution of the object. Projections in a given direction are then computed by rotating the matrix and summing values along the axis perpendicular to that direction. For example, for the x-projection, we sum the scores for each x-coordinate (with segmentation scores normalized as 1/mask size, and cross-attention scores as values in the interval of 0 to 1), yielding the projected distribution on which PoS is directly applied. Full implementation details are provided in the supplementary code for PSE and PSG.
>
> Regarding the choice of layer, we follow prior work [1,2] and use the cross-attention map from the layer closest to the middle of the U-Net. This layer has been shown to exhibit the strongest correlation with semantic attributes of the generated image [2]. We further conducted an ablation on effectiveness of PSG using different layers of the U-Net, mentioned in the following table (Table 12). We observed similar trends as suggested by [2], which resulted in our particular choice of the layer.
>
> | Metric | Without PSG | 32×32 Maps PSG | 64×64 Maps PSG |
> |--------|-------------|----------------|----------------|
> | VISOR  | 44.0        | **74.4**       | 48.2           |
> | PSE    | 21.0        | **62.0**       | 32.0           |
>
>
> 2- **Multi-Object, Multi-Relation**
>
> Our paper evaluates exactly these cases on both the generation and evaluation sides:
>
> **Generalizability of PSG to multiple objects/relations:**
>
> - Qualitative evidence: Figure 1 contains prompts with several relations among multiple entities, and PSG maintains correct layouts across them. Additional examples of combined (e.g., top-left) relations are shown in Appendix B.3 (Figure 18), illustrating that PSG composes multiple directional constraints reliably.
>
> - Quantitative evidence on multi-object prompts: We report results on HRS-Benchmark, whose prompts establish relationships among multiple objects within each prompt. PSG improves spatial accuracy over strong baselines on this benchmark (Table 6), demonstrating effectiveness specifically in the multi-object, multi-relation setting.
>
> - Mechanism: Gradient-based PSG operates on cross-attention maps and sums losses for all the relation pairs, enabling composition; search-based PSG evaluates complete images, so it naturally scales with the number of objects/relations expressed in the prompt.
> Moreover, in the following table (Table 15), we used PSG on top of SDXL to generate images using multi-object multi-relation prompts. Even though the task becomes increasingly difficult and the performance of PSG would depend on the limited capacity of the backbone model, we observe noticeable improvement upon the baseline.
>
> | # Objects | Model        | PSE   | VISOR |
> |-------------|--------------|-------|-------|
> | 3           | SDXL         | 30.8  | 57.0  |
> |  3           | SDXL+PSG     | **63.8** | **72.5** |
> | 4           | SDXL         | 31.5  | 54.0  |
> |   4          | SDXL+PSG     | **68.4** | **58.0** |
>
>
> **Robustness of the metric (PSE) for multi-object scenes:**
>
> - Design: PSE is defined per relation between object pairs using Projected-PoS; complex directions (e.g., top-right, between) arise by combining 1D projections (Appendix B.4). This lets us evaluate each stated relation in a multi-object prompt and aggregate per benchmark protocol.
>
> - Detector robustness: We further demonstrate robustness across different detection/segmentation models and under mask corruptions (Appendix A.3; Tables 10 and 11), providing evidence that the metric can remain stable even in dense scenes. While we employed state-of-the-art detection models in our experiments, our method is detector-agnostic and can readily benefit from future models that may achieve stronger multi-object performance.

---

> > ### Author Response · Authors · 2025-09-26
> > **Response to reviewer 4mMa [Part 2]**
> >
> > 3- **Behind and Covered**
> >
> > Our framework already models certain 3D spatial relations and evaluates them empirically:
> > - Behind: We explicitly test depth-ordering relations. For example, Figure 19 includes prompts such as “A chair behind a desk” and “A dog hidden by a frog”, where hidden, as in T2I-CompBench, is treated as equivalent to behind. PSG consistently improves alignment in these 3D cases. Depth ordering is handled using monocular depth estimation to obtain z-aware masks, which are then used in computing the PoS projection (see Section 3).
> > - Covered: This relation is more challenging, as its meaning is often underspecified and can vary by user interpretation, making it difficult to design a general, rule-based evaluation. Nevertheless, our framework already incorporates both distance guidance (Appendix B.4) and depth guidance (Section 3). In principle, combining these mechanisms provides a natural way to model “Covered.” However, given the subjectivity of the concept, such a formulation would probably be less general than our treatment of directional relations.
> >
> > 4- **Image Editing**
> >
> > Image editing methods of various kinds can also benefit from our approach. As discussed in Section 3.2 of [3], during diffusion generation one can preserve the attention values of the original image and apply an edit function at each step to smoothly modify the output. This edit can take the form of reweighing, as in the original work [3], or continuous latent optimization. Building on Section C.1 of A&E [1], PoS provides another avenue for editing. For gradient-based PSG, we can apply it directly as the editing function, iteratively refining the cross-attention maps to achieve more accurate spatial relationships. For search-based PSG, recent work has emphasized the use of external verifiers for editing in text-to-image models [4], using both continuous and discrete search. In this context, PSE can complement verifiers such as DreamSim [5], which target concept preservation, by rewarding images that retain content while achieving new positional relationships. Since PoS provides a continuous score interpolating between incorrect and correct relations, it offers a non-sparse reward signal that is expected to facilitate smoother and more effective editing.
> >
> > 5- **PSG during training process**
> >
> > Our primary contributions are twofold: (i) introducing a verifier for spatial relationships, and (ii) demonstrating how this verifier can be exploited with additional test-time computation to enhance spatial performance. Although our method is applied at inference, the continuity of the proposed loss indicates that, in principle, it could also be incorporated into the training process. However, prior works (e.g. [1]) have shown that latent guidance can be effective as a test-time intervention rather than as a training objective. We therefore leave for future work the exploration of how verifier-based guidance, currently employed only at inference, might be adapted into a novel training paradigm for diffusion models.
> >
> > [1] Chefer et al., Attend-and-Excite: Attention-Based Semantic Guidance for Text-to-Image Diffusion Models, ACM, 2023
> >
> > [2] Surkov et al., Unpacking SDXL Turbo: Interpreting Text-to-Image Models with Sparse Autoencoders, Mechanistic Interpretability for Vision at CVPR 2025, 2025
> >
> > [3] Hertz et al., Prompt-to-Prompt Image Editing with Cross-Attention Control, ICLR, 2023
> >
> > [4] Beyer et al., Highly Compressed Tokenizer Can Generate Without Training, arXiv preprint, 2025
> >
> > [5] Fu et al., DreamSim: Learning New Dimensions of Human Visual Similarity Using Synthetic Data, NeurIPS, 2023

---

### Review · Reviewer_fbcu · 2025-08-23

**Summary Of Contributions:**

The paper proposes a probabilistic formulation of spatial relations between objects in text-to-image (T2I) generation using the Probability of Superiority (PoS). Specifically, the authors introduce (i) PSE, an evaluation metric that compares projected distributions (from masks and optional depth) to score 2D/3D relations, and (ii) PSG, an inference-time guidance method that improves spatial alignment either by gradient nudging on cross-attention maps or by Best-of-N selection using PSE as a reward. They further present OPSE, a UCB bandit procedure to identify the best model with fewer samples. Empirically, PSE correlates better with human judgments than center-based metrics, and PSG improves spatial benchmarks (VISOR, T2I-CompBench, HRS) with modest quality trade-offs.

**Audience:**

Yes

**Claims And Evidence:**

Yes

**Requested Changes:**

There is a typo in the last figure in Figure 1 "A computer in fornt of a cow."

**Strengths And Weaknesses:**

Strengths:
1. The paper is well-written and easy to follow.
2. Modelling the spatial relationships between objects in generated image using probability is intuitively more robust than using center points. The human study also shows higher correlation for PSE than other metrics, such as VISOR, T2I-CompBench, etc.
3. The proposed PoS-based generation is a simple and straightforward approach to enhance generation with specified spatial configurations.

Weaknesses:
1. What is the inter-rater agreement for the human study and how are the human rater results aggregated?
2. Why only a specific PSG method is chosen for each benchmark in Section 4?  E.g. only gradient-based PSG is evaluated in Table 4-6. Is it because search-based PSG cannot bring significant benefits on these benchmarks?
3. For the evaluations for PoS-based generation,  is it possible to also employ gradient-based or search-based method for image generation using other metrics, such as VISOR? Comparing the generation results with PSE-guided generation would clearly show the advantage of PSE.

---

> ### Author Response · Authors · 2025-09-26
> **Response to reviewer fbcu**
>
> We thank the reviewer for the positive feedback. We are glad that the paper was found clear and easy to follow, that PSE’s probabilistic modeling was recognized as more robust than center-point approaches, and that its stronger correlation with human judgment was noted. We also appreciate the acknowledgment of the simplicity and effectiveness of our PoS-based generation method. Below we provide detailed responses to the raised concerns.
>
> 1- **Human-rater details**
>
> As described in Section 4.1 (“Alignment to Human Judgment”), three independent evaluators assessed 2,000 image–prompt pairs using a custom interface. Each rater selected one of three labels: (1) object missing, (2) object present but relationally incorrect, or (3) correctly aligned.
> This three-level design was intended to minimize ambiguity and reduce subjectivity. By distinguishing between “object missing” and “relationally incorrect,” we enabled more precise judgments, which improved consistency across annotators.
> For result aggregation, we computed separate scores for object accuracy and relational accuracy. In the final evaluation, type (3) responses were converted to 1, while types (1) and (2) were converted to 0. The aggregated score for each prompt–image pair was then obtained by averaging across the three raters.
>
> 2- **The comparison of two PSG Methods**
>
> To clarify, both PSG variants, gradient-based (continuous optimization) and search-based, are applicable across all mentioned benchmarks. Our decision to report a specific variant in each case was motivated by methodological suitability rather than inherent performance limitations.
> For 2D spatial relations, gradient-based PSG is the natural (more efficient, refer to Figure 10) choice because cross-attention maps provide reliable 2D localization cues during generation. This is why Tables 4–6 focus on the gradient-based variant.
> For 3D spatial relations, however, cross-attention maps do not directly encode depth information. In this setting, search-based PSG, which evaluates fully generated outputs with depth-aware scoring, is more appropriate. This explains our emphasis on search-based PSG in Table 7.
> We emphasize that search-based PSG can also be applied to 2D relations and would yield further gains given higher computational budgets (as discussed in Appendix B.1), similar to other test-time scaling strategies. Thus, our presentation reflects a pragmatic division: gradient-based PSG when intermediate cross-attention information is sufficient, and search-based PSG when such information is unavailable, rather than a limitation of the method itself. Figure 10 provides a direct comparison of these two approaches under varying compute.
>
> 3- **Guidance using alternative method (VISOR)**
>
> Applying gradient-based optimization with VISOR is not feasible, as VISOR is non-continuous and therefore not differentiable. This prevents it from being used in a gradient-based guiding framework.
> Regarding search-based methods, we note that a key advantage of PSE is that it provides a continuous score indicating the degree to which a spatial relationship is satisfied. This property is particularly important in multi-step search algorithms, where intermediate feedback is needed to effectively guide the search. By contrast, VISOR only provides binary judgments, which limit its usefulness in such iterative settings.
> However, to analyze the best-of-N generation, we include in the revised version an additional experiment comparing VISOR versus PSE-guided search. Since evaluating generations with the same metric used for guidance may lead to reward hacking, we rely on human evaluation as an impartial reference. Results in the following table (Table 17 of the paper) confirm that PSE-guided PSG achieves higher alignment with human judgment.
> | Method             | VISOR ↑ | PSE ↑ | Human ↑ |
> |--------------------|---------|-------|---------|
> | Search-Based-VISOR | **92.0** | 33.8  | 22.0    |
> | Search-Based-PSG   | 72.0    | **77.5** | **61.0** |
>
> 4- **Caption Typo**
>
> We thank the reviewer for noticing this typo. In the revised version of the manuscript, the mentioned prompt in Figure 1 has been corrected from “in fornt” to “in front.”

---

### Review · Reviewer_Qw6y · 2025-09-13

**Summary Of Contributions:**

This paper proposes a probabilistic framework to model the relative spatial positioning of different objects in a scene. The authors argues that the current text-to-image generation models struggles to accurately follow input prompts to reflect the specified spatial configuration between different objects, and the existing evaluation metrics (VISOR, T2I-CompBench etc) can not effectively penalize such deficiencies in the generated images. In this work, the authors first proposes a PoS (Probability of Superiority) - based evaluation metric, PSE, which is designed to access the alignment of objects in 2D and 3D scenes generated by T2I models. Next, based on PSE, the authors propose PoS-based Generation (PSG), an inference-time method that improves spatial alignment without requiring extra fine-tuning or additional input. Lastly, this paper also defines online PSE (OPSE), an online version of the PSE metric to reliably detect the superior model using only a limited number of samples. The experiments show that PSG improves the ability of T2I models to preserve stronger spatial relationship, and PSE metric exhibits stronger alignment with human judgment compared to traditional center-based metrics.

**Audience:**

Yes

**Claims And Evidence:**

Yes

**Requested Changes:**

Please refer to the major weaknesses section.

**Strengths And Weaknesses:**

**Strengths**

The paper is generally well-written and easy-to-understand. The authors systematically define the limitations of existing evaluation metrics like VISOR, and builds upon a simple solution to better model the spatial relationship between objects. The proposed metric (PSE) also correlates to human judgement well (better than VISOR, and other existing metrics) which shows the effectiveness of PSE. For PSG, both the gradient-based and search-based methods make sense, and overall the experimental results supports the claims. The qualitative samples are adequate, and depicts the usefulness of the proposed metric.

**Major Weaknesses**

(1) As the authors say in A.5., the success of the proposed metric entirely depends on the accuracy of the segmentation and depth estimation model. For densely populated scenes, Grounding DINO may fail in locating complex referring text prompt (such as "object in the far right which in not in the bottom"), and the metric will inherently incur noise. I would suggest the authors in reporting such failure case and consider providing an ablation with different segmentation and depth estimation modules.

(2) Though the authors claim in B.4. that the concept of relative distance ('close' vs 'far') can be modeled by PoS by incorporating a constant 'c', the value for 'c' can be different based on the context. For example, 'far' in a zoomed out image would require a lower value of 'c' compared to a zoomed in view. Though I agree that the same issue is present in previous metrics like VISOR, PSE does not solve it. I would also suggest the authors to provide ablation on the value of 'c', and provide some visualization for them.

(3) The search-based PSG method requires generating N samples per prompt which can be extremely expensive for large T2I models. Moreover, choosing a higher value of N is important to achieve a substantial gain. The paper does not provide any concrete analysis of latency for PSG.

(4) I would also be interested to know the effect of PSD for T2V models (though out of the scope for this particular paper, but is interesting).

 **Minor Weakness**

(1) The abstract mentions 'PSG employs a Part-of-Speech PoS-based reward function ...'; what is Part-of-Speech?


Overall, the paper provides a good contribution with adequate experiments and supports most of its claims effectively by pointing out limitations in previous evaluation metrics.

---

> ### Author Response · Authors · 2025-09-26
> **Response to reviewer Qw6y [Part 1]**
>
> We thank reviewer Qw6y for the thoughtful and encouraging feedback. We are pleased that the reviewer found the paper well-written and easy to follow, and appreciated our systematic analysis of the limitations of existing metrics such as VISOR. We are also glad that the effectiveness of our proposed PSE metric was recognized, along with the soundness of our PSG methods. Please find our responses to the mentioned concerns below.
>
> 1- **Exterior Model Robustness**
>
> We acknowledge that evaluation metrics such as PSE or VISOR, which rely on external detectors, inevitably inherit the limitations of those models. However, the detector-agnostic design of PSE ensures that it can directly benefit from future advances in segmentation and depth estimation. To assess robustness, we conducted an extensive study in Appendix A.3 (Robustness of Detector). While we selected Grounded SAM due to its open-vocabulary capability and superior performance over other detectors, we also compared our method using alternative state-of-the-art detection models (CLIPSeg, Mask2Former), as reported in Table 11.
> Furthermore, to evaluate the effect of segmentation errors, we introduced controlled mask corruptions (Table 10) and found that PSE remains stable under such perturbations. This robustness helps explain why PSG, which operates on imperfect cross-attention maps rather than perfect masks, can still reliably guide the generation.
> Overall, we agree that highly complex referring expressions remain a challenge for current detection models. We regard this as an open problem, but one where PSE already demonstrates resilience (Appendix A.3) and stands to further improve as detection and depth estimation methods advance.
>
>
> 2- **Distance-based PoS**
>
> We agree with the reviewer that the notion of “closeness” is inherently subjective and context-dependent, as the value of c can vary with factors such as image scale or viewpoint. Our intention in introducing this direction was to provide a mechanism, absent in prior works such as VISOR, that enables users to more accurately capture relative distance relations when desired. The value of c can be treated as a tunable hyperparameter, allowing adaptation to different scenarios, and it may also be heuristically defined based on attributes such as the bounding box size of the objects.
> In line with the reviewer’s suggestion, we have added a new visualization (Figure 20) in the Appendix of the revised paper, which illustrates the effect of distance guidance.
>
> 3- **Latency of PSG**
>
> To clarify, PSG was introduced in two complementary forms: gradient-based and search-based. The gradient-based variant provides a practical option for improving model performance with only a modest additional compute cost. As shown in Figure 10 of the Appendix and in our ablation on hyperparameters, the achievable gain is bounded by the best-performing timestep (10 in our case). Since positional relations typically require only a few early timesteps to guide the generation, the gradient-based approach is generally faster than optimization for other aspects such as attribute binding [1], where 20–30 timesteps are more effective.
> In contrast, the search-based variant is designed as a test-time scaling method, intended for scenarios with access to higher computational budgets. This formulation aligns with recent work on scaling compute at inference, including best-of-N (BoN) strategies, which have been shown to be effective for diffusion models [2]. As illustrated in Figure 10, accuracy improves consistently as additional compute is allocated (with time on the x-axis representing compute). Importantly, the search-based PSG can also be parallelized, enabling faster generation by distributing computation across multiple GPUs rather than increasing latency per sample.

---

> > ### Author Response · Authors · 2025-09-26
> > **Response to reviewer Qw6y [Part 2]**
> >
> > 4- **T2V models**
> >
> > We appreciate the reviewer’s interest in exploring the applicability of PSE to text-to-video (T2V) models. Although this direction is beyond the immediate scope of our paper, PSE is inherently frame-based, and thus can be directly applied to individual video frames to assess spatial accuracy. As demonstrated in Section 4.1, PSE correlates strongly with human ratings, making it a reliable verifier of positional accuracy. This property suggests that PSE could naturally extend to T2V settings by serving as a criterion for ranking or pruning candidate frames. For example, in works such as Video-T1 [3], PSE could help improve spatial alignment by continuously scoring frames and guiding the selection process. We view this as a promising avenue for future research.
> >
> > 5- **Abstract typo**
> >
> > The phrase “Part-of-Speech PoS-based reward function” in the abstract was a typographical error. It should read simply as “PoS-based reward function.” We apologize for the confusion, and this has been corrected in the revised manuscript.
> >
> >
> > [1] Chefer et al., Attend-and-Excite: Attention-Based Semantic Guidance for Text-to-Image Diffusion Models, ACM, 2023
> >
> > [2] Ma et al., Inference-Time Scaling for Diffusion Models Beyond Scaling Denoising Steps, arXiv preprint, 2025
> >
> > [3] Liu et al., Video-T1: Test-Time Scaling for Video Generation, arXiv preprint, 2025

---

### Author Response · Authors · 2025-09-26

We thank the reviewers for their constructive and thoughtful feedback. An updated version of the paper has been submitted, with major revisions highlighted in blue for clarity.

---

### Decision · Action_Editor_t1E2 · 2025-12-01

**Recommendation:** Accept with minor revision

**Additional Comments:**

Please incorporate changes made in the rebuttal and address the additional concerns from reviewers.

**Audience:**

Yes

**Audience Explanation:**

Spatial alignment in text2image modeling is an important issue that the generative modeling community care about. Therefore this work should appeal to a large population of the TMLR audience.

**Claims And Evidence:**

Yes

**Claims Explanation:**

As agreed by all reviewers, the proposed PoS based metric appears to be effective in detecting the spatial alignment in the generated images and using it as a reward does indeed improve the sampling quality.

---

> ### Author Response · Authors · 2025-12-04
> **Camera-Ready Submission Note**
>
> We have uploaded the camera-ready submission, incorporating the requested changes and the clarifications discussed in the rebuttal. We appreciate the constructive feedback and thank the reviewers and the action editor for their comments.